# Lattice partition recovery with dyadic CART

**Oscar Hernan Madrid Padilla**
Department of Statistics
University California, Los Angeles
oscar.madrid@stat.ucla.edu

**Yi Yu**
Department of Statistics
University of Warwick
yi.yu.2@warwick.ac.uk

**Alessandro Rinaldo**
Department of Statistics & Data Science
Carnegie Mellon University
arinaldo@cmu.edu

## Abstract

We study piece-wise constant signals corrupted by additive Gaussian noise over a $d$-dimensional lattice. Data of this form naturally arise in a host of applications, and the tasks of signal detection or testing, de-noising and estimation have been studied extensively in the statistical and signal processing literature. In this paper we consider instead the problem of partition recovery, i.e. of estimating the partition of the lattice induced by the constancy regions of the unknown signal, using the computationally-efficient dyadic classification and regression tree (DCART) methodology proposed by [14]. We prove that, under appropriate regularity conditions on the shape of the partition elements, a DCART-based procedure consistently estimates the underlying partition at a rate of order $\sigma^2 k^* \log(N)/\kappa^2$, where $k^*$ is the minimal number of rectangular sub-graphs obtained using recursive dyadic partitions supporting the signal partition, $\sigma^2$ is the noise variance, $\kappa$ is the minimal magnitude of the signal difference among contiguous elements of the partition and $N$ is the size of the lattice. Furthermore, under stronger assumptions, our method attains a sharper estimation error of order $\sigma^2 \log(N)/\kappa^2$, independent of $k^*$, which we show to be minimax rate optimal. Our theoretical guarantees further extend to the partition estimator based on the optimal regression tree estimator (ORT) of [12] and to the one obtained through an NP-hard exhaustive search method. We corroborate our theoretical findings and the effectiveness of DCART for partition recovery in simulations.

## 1 Introduction

Suppose we observe a noisy realization of a structured, piece-wise constant signal supported over a $d$-dimensional square lattice (or grid graph) $L_{d,n} = \{1, \ldots, n\}^d$. Data that can be modeled in this manner arise in several application areas, including in satellite imagery [e.g. 30, 37], computer vision [e.g. 6, 38], medical imaging [e.g. 26, 23], and neuroscience [e.g. 16, 32]. Our goal is to estimate the constancy regions of the underlying signal. Specifically, we assume that the data $y \in \mathbb{R}^{L_{d,n}}$ are such that, for each coordinate $i \in L_{d,n}$,

$$y_i = \theta_i^* + \epsilon_i, \tag{1}$$

where $(\epsilon_i, i \in L_{d,n})$ are i.i.d. $\mathcal{N}(0, \sigma^2)$ noise variables and the unknown signal $\theta^* \in \mathbb{R}^{L_{d,n}}$ is assumed to be piece-wise constant over an unknown rectangular partition of $L_{d,n}$. We define a subset $R \subset L_{d,n}$ to be a *rectangle* if $R = \prod_{i=1}^d [a_i, b_i]$, where $[a, b] = \{j \in \mathbb{Z} : a \leq j \leq b\}$, $a, b \in \mathbb{Z}$. A *rectangular partition* of $L_{d,n}$, $\mathcal{P}$, is a collection of disjoint rectangles $\{R_l\} \subset L_{d,n}$, satisfying $\cup_{R \in \mathcal{P}} R = L_{d,n}$. To each vector in $L_{d,n}$, there corresponds a (possibly trivial) rectangular partition.

35th Conference on Neural Information Processing Systems (NeurIPS 2021).

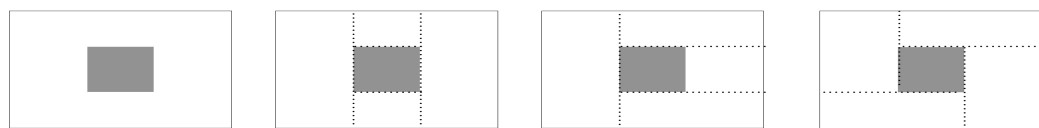

Figure 1: Rectangular partitions associated to a vector are not necessarily unique.

**Definition 1** *A rectangular partition associated with a vector $\theta \in \mathbb{R}^{L_{d,n}}$ is a rectangular partition $\{R_l\}_{l \in [1,k]}$ of $L_{d,n}$, such that $\theta$ takes on constant values over each $R_l$. For a vector $\theta \in \mathbb{R}^{L_{d,n}}$, we let $k(\theta)$ be the smallest positive integer such that there exists a rectangular partition with $k(\theta)$ elements and associated with $\theta$.*

In this paper, we are interested in recovering a rectangular partition associated with the signal $\theta^*$ in (1). A complication immediately arises when $d \geq 2$: the rectangular partition associated with a given $\theta$ is not necessarily unique. This fact is illustrated in Figure 1, where the left-most plot depicts the lattice supported vector $\theta$, consisting of a rectangle of elevated value (in grey) against a background (in white). For such $\theta$, we show three possible rectangular partitions, each of which comprised of five rectangles (the second, third and fourth plots). In fact, the partition recovery problem is well defined, as long as we consider coarser partitions comprised by unions of adjacent rectangles instead of individual rectangles: see Definition 2 below for details. We remark that this issue does not occur in the univariate ($d = 1$) case, for which the partition recovery task has been thoroughly studied in the change point literature; see section 1.3 below. Thus, we assume that $d \geq 2$.

For the purpose of estimating the rectangular partition associated with $\theta^*$ (or, more precisely, its unique coarsening as formalized in Definition 2), we resort to the dyadic classification and regression tree (DCART) algorithm of [14]. This is a polynomial-time decision-tree-based algorithm developed for de-noising purposes for signals over lattices, and is a variant of the classification and regression trees (CART) [7]. See Section 1.1 below for a description of DCART. The optimal regression trees (ORT) estimator, recently proposed in [12], further builds upon DCART and delivers sharp theoretical guarantees for signal estimation while retaining good computational properties – though we should mention that in our experiments we have found DCART to be significantly faster. Both DCART and its more sophisticated version ORT can be seen as approximations to the NP-hard estimator

$$\theta_1 = \arg\min_{\theta \in \mathbb{R}^{L_{d,n}}} \left\{ 2^{-1} \|y - \theta\|^2 + \lambda k(\theta) \right\}, \tag{2}$$

where $k(\theta)$ is given in Definition 1, $\| \cdot \|$ is the vector (or Euclidean) $\ell_2$-norm and $\lambda > 0$ a tuning parameter. DCART modifies the above, impractical optimization problem by restricting only to dyadic rectangular partitions. This leads to significant gains in computational efficiency without sacrificing on the statistical performance. Indeed, decision-tree-based algorithms have been shown to be optimal under various settings for the purpose of signal estimation; see [12, 20]. In this paper, we further demonstrate their effectiveness for the different task of partition recovery. In particular, we show how simple modifications of the DCART (or ORT) estimator yield practicable procedures for partition recovery with good theoretical guarantees and derive novel localization rates.

Note that, there is a wide array of applications focusing on detecting the regions rather than estimating the background signals, especially in surveillance and environment monitoring. Our work is motivated by all the applications/problems considered in the large literature on biclustering, where the underlying signal is assumed to be piecewise constant. Estimating the boundary of the partition is the most refined and difficult task in these settings. Thus, any of the many scenarios in which biclustering is relevant can be used to motive our task. An analogous observation holds also for the more general problem of identifying an anomalous cluster (sub-graph) in a network, a problem that has been tackled (for testing purposes only) by [4], the reference therein provide numerous examples of applications. On a high-level, the relationship between the partition and signal recoveries can be thought of the relationship between the estimation consistency and support consistency in a high-dimensional linear regression problems. They can be done by almost identical algorithms but the theoretical results rely on different sets of conditions.

The paper is organized as follows. In the rest of this section we formalize the problem settings and the task of partition recovery, and describe the DCART procedure. We further summarize our main findings and discuss related literature. Section 2 contains our main results about one- and

two-sided consistency of DCART and its modification. In Section 2.3 we derive a minimax lower bound stemming from the case of one rectangular region of elevated signal against pure background. Illustrative simulations corroborating our findings can be found in Section 3. The Supplementary Material contains the proofs.

**Notation**

We set $N = n^d$, the size of the lattice $L_{d,n}$, where we recall that $d \geq 2$ is assumed fixed throughout. For any integer $m \in \mathbb{N}^*$, let $[m] = [1, m]$. Given a rectangular partition $\Pi$ of $L_{d,n}$, let $S(\Pi)$ be the linear subspace of $\mathbb{R}^{L_{d,n}}$ consisting of vectors with constant values on each rectangle in $\Pi$ and let $O_{S(\Pi)}(\cdot)$ be the orthogonal projection onto $S(\Pi)$. For any $R \subset L_{d,n}$ and $\theta \in \mathbb{R}^{L_{d,n}}$, let $\bar{\theta}_R = |R|^{-1} \sum_{i \in R} \theta_i$, where $|\cdot|$ is the cardinality of a set. Two rectangles $R_1, R_2 \in \Pi$ are said to be *adjacent* if there exists $l \in [d]$ such that $R_1$ and $R_2$ share a boundary along $e_l$ and one is a subset of the other in the hyperplane defined by $e_l$, the $l$th standard basis vector in $\mathbb{R}^d$. See Definition S1 for a rigorous definition. This concept of adjacency is specifically tailored to – and in fact only valid for – dyadic (and hierarchical, in the sense specified by [12]) rectangular partitions, which are most relevant for this paper. For any subsets $A, B \subset L_{d,n}$, define $\mathrm{dist}(A, B) = \min_{a \in A, b \in B} \|a - b\|$. Throughout this paper, we will use the $\ell_2$-norm as the vector norm.

## 1.1   Problem setup

We begin by introducing two key parameters for the model specified in (1) and a well-defined notion of rectangular partition induced by $\theta^*$.

**Definition 2 (Model parameters, induced partitions)** *Let $\theta^*$ as in (1) and $\{R_j^*\}_{j \in [m]}$ be a rectangular partition of $L_{d,n}$ associated with $\theta^*$. Consider the graph $G^* = (E^*, V^*)$, where $V^* = [m]$ and $E^* = \{(i,j) : \bar{\theta}_{R_i^*} = \bar{\theta}_{R_j^*}, R_i^* \text{ and } R_j^* \text{ are adjacent}\}$. Let $\{C_l^*\}_{l \in [L]}$ be all connected components of $G^*$ and define $\Lambda^* = \Lambda^*(\theta^*) = \{\cup_{j \in C_1^*} R_j^*, \ldots, \cup_{j \in C_L^*} R_j^*\}$ as the partition (not necessarily rectangular) induced by $\theta^*$. We say that the union of rectangles $\cup_{j \in C_s^*} R_j^*$ and $\cup_{j \in C_t^*} R_j^*$, $s, t \in [L]$, $s \neq t$, are adjacent, if and only if there exists $(i,j) \in C_s^* \times C_t^*$ such that $R_i^*$ and $R_j^*$ are adjacent.*

*Let $\kappa$ and $\Delta$ be the minimum jump size and minimal rectangle size, respectively, formally defined as*

$$\kappa = \min_{a \in A, b \in B, \, A, B \in \Lambda^*, \, \theta_a^* \neq \theta_b^*} |\theta_a^* - \theta_b^*| \quad and \quad \Delta = \min_{j \in [m]} |R_j^*|.$$

It is important to emphasize the difference between a partition associated with $\theta^*$, as described in Definition 1, which may not be unique, and the partition $\Lambda^*$ induced by $\theta^*$ of Definition 2, which is instead unique and thus describes a well-defined functional of $\theta^*$. The parameters $\kappa$ and $\Delta$ capture two complementary aspects of the intrinsic difficulty of the problem of estimating $\Lambda^*$; intuitively, one would expect the partition recovery task to be more difficult when $\kappa$ and $\Delta$ are small (and $\sigma$ is large). Below, we will prove rigorously that this intuition is indeed correct. When $d = 1$, both parameters, along with $\sigma$, have in fact been shown to fully characterize the change point localization task: see, e.g., [36, 35].

The partition recovery task can therefore be formulated as that of constructing an estimator $\widehat{\Lambda}$ of $\Lambda^*$, the induced partition of $\theta^*$, such that, as the sample size $N$ grows unbounded and with probability tending to one,

$$|\widehat{\Lambda}| = |\Lambda^*| \quad \text{and} \quad \Delta^{-1} d_{\mathrm{Haus}}(\widehat{\Lambda}, \Lambda^*) = \Delta^{-1} \max_{A \in \Lambda^*} \min_{B \in \widehat{\Lambda}} |A \triangle B| \to 0, \tag{3}$$

where $A \triangle B$ is the symmetric difference between $A$ and $B$. We refer to $d_{\mathrm{Haus}}(\widehat{\Lambda}, \Lambda^*)$ as the localization error for the partition recovery problem.

**The dyadic classification and regression trees (DCART) estimator.** In order to produce a computationally efficient estimator of $\Lambda^*$ satisfying the consistency requirements (3), we deploy the DCART procedure [14], which can be viewed as an approximate solution to the problem in (2). Instead of optimizing over all vectors in $\mathbb{R}^{L_{d,n}}$, DCART minimizes the objective function only over vectors associated with a *dyadic rectangular partition*, which is defined as follows. Let $R = \prod_{i \in [d]} [a_i, b_i] \subset L_{d,n}$ be a rectangle. A *dyadic split* of $R$ chooses a coordinate $j \in [d]$, $l$ the middle point of $[a_j, b_j]$, and

splits $R$ into

$$R_1 = \prod_{i \in [j-1]} [a_i, b_i] \times [a_j, l] \times \prod_{i \in [j+1,d]} [a_i, b_i] \text{ and } R_2 = \prod_{i \in [j-1]} [a_i, b_i] \times [l+1, b_j] \times \prod_{i \in [j+1,d]} [a_i, b_i],$$

with $[0] = [c_2, c_1] = \emptyset$, $c_2 > c_1$. Assuming that $n$ is a power of 2, starting from $L_{d,n}$ itself, we proceed iteratively as follows. Given the partition $\{R_u\}_{u \in [k]}$, one chooses a rectangle $R_u$ and performs a dyadic split on $R_u$ that leads to the largest reduction in the objective function. Any partition constructed through a sequence of such steps is called a dyadic rectangular partition. With a pre-specified $\lambda > 0$, the DCART estimator is

$$\tilde{\theta} = O_{S(\widetilde{\Pi})}(y), \text{ where } \widetilde{\Pi} \in \underset{\Pi \in \mathcal{P}_{\text{dyadic},d,n}}{\arg\min} \left\{ 2^{-1}\|y - O_{S(\Pi)}(y)\|^2 + \lambda|\Pi| \right\}, \tag{4}$$

where $\mathcal{P}_{\text{dyadic},d,n}$ is the set of all dyadic rectangular partitions of $L_{d,n}$. As shown in [14] and [12], the DCART estimator can be obtained via dynamic programming with a computational cost of $O(N)$. Given any solution to (4), a natural (though, as we will see, sub-optimal) estimator of the induced partition of $\theta^*$ is $\widetilde{\Pi}$, the partition associated with the resulting DCART estimator $\tilde{\theta}$. Importantly, by the property of DCART, and using the fact that the Gaussian errors have a Lebesgue density, $\widetilde{\Pi}$ is in fact a dyadic-rectangular partition and is unique with probability one, and thus the resulting estimator is well-defined. (Equivalently, the partition associated with $\tilde{\theta}$ and the one induced by $\tilde{\theta}$ coincide.)

## 1.2 Summary of our results

We briefly summarize the contributions made in this paper.

**One-sided consistency of DCART.** Though DCART is known to be a minimax rate-optimal estimator of $\theta^*$ [12], for the task of partition recovery its associated partition $\widetilde{\Pi}$ has sub-optimal performance. Indeed, due to the nature of the procedure, it is easy to construct cases in which the DCART over-partitions. See Figure 2. In these situations, DCART falls short with respect to the target conditions for consistency described in (3). Nonetheless, it is possible to prove a weaker one-sided consistency guarantee, in the sense that every resulting DCART rectangle is almost constant. In detail, let $\mathcal{R} = \{R_l\}_{l \in [\tilde{m}]}$ be the rectangular partition defined by $\widetilde{\Pi}$ in (4). Then, we show in Section 2.1 that, for any $R_i \in \mathcal{R}$, there exists $S_i \subset R_i$ such that $\theta_t^* = \theta_u^*$, $u, t \in S_i$, and $\sum_{i \in [\tilde{m}]} |R_i \backslash S_i| \lesssim \kappa^{-2}\sigma^2 k_{\text{dyad}}(\theta^*) \log(N)$. Throughout, the quantity $k_{\text{dyad}}(\theta^*)$ refers to the smallest positive integer $k$ such that there is a $k$-dyadic-rectangular-partition of $L_{d,n}$ associated with $\theta^*$.

**Two-sided consistency of DCART: A two-step estimator.** In order to resolve the unavoidable over-partitioning issue with the naive DCART partition estimator and in order to prevent the occurrence of spurious clusters, we develop a more sophisticated two-step procedure. In the first step we use a variant of DCART that discourages the creation of rectangles of small volumes. In the second step, we apply a pruning algorithm merging rectangles when their values are similar and the rectangles are not far apart. With probability tending to one as $N \to \infty$, the final output $\widehat{\Lambda}$ satisfies (3) with $d_{\text{Haus}}(\widehat{\Lambda}, \Lambda^*) \leq \kappa^{-2}\sigma^2 k_{\text{dyad}}(\theta^*) \log(N)$. This result is the first of its kind in the setting of lattice with arbitrary dimension $d \geq 2$. This is shown in Section 2.2.

**Optimality: A regular boundary case.** In Section 2.3, we consider the special case in which, for each rectangle in the rectangular partitions induced by $\theta^*$ only has $O(1)$-many rectangles within distance of order $\sigma^2 \kappa^{-2} k_{\text{dyad}}(\theta^*) \log(N)$. While more restrictive than the scenarios we study in Sections 2.1 and 2.2, this setting is broader than the ones adopted in the cluster detection literature [e.g. 2, 1]. In this case, with probability approaching one as $N \to \infty$, the estimator $\widehat{\Lambda}$ satisfies (3) and $d_{\text{Haus}}(\widehat{\Lambda}, \Lambda^*) \leq \kappa^{-2}\sigma^2 \log(N)$. This error rate is shown to be minimax optimal, with a supporting minimax lower bound result in Proposition 3.

## 1.3 Related and relevant literature

The problem at hand is closely related to several recent research threads involving detection and estimation of a structured signal. When $d = 2$, our settings can be viewed as a generalization of those used for the purpose of biclustering, i.e. detection and estimation of sub-matrices. Though

relatively recent, the literature on this topic is extensive, and the problem has been largely solved, both theoretically and methodologically. See, e.g., [28], [21], [10], [25], [31], [24], [5], [11], [9], [17], [18], [13] and [28].

In the more general settings postulating a structured signal supported over a graph (including the grid graph), sharp results for the detection problem of testing the existence of a sub-graph or cluster in which the signal is different from the background are available in the literature: see, [3], [2], [1]. Concerning the estimation problem, [34], [29], [12], [15] and others, focused on de-noising the data and upper-bounding $\|\hat{\theta} - \theta^*\|_*$, where $\hat{\theta}$ is an estimator of $\theta^*$ and $\|\cdot\|_*$ is some vector norm. In yet another stream of work [e.g. 19, 8, 22] concerned with empirical risk minimization, the problem is usually formulated as identifying a single subset. More discussions can be found in Section S1.

What sets our contributions apart from those in the literature referenced above, which have primarily targeted detection and signal estimation, is the focus on the arguably different task of partition recovery. As a result, the estimation bounds we obtain are, to the best of our knowledge, novel as they do not stem directly from the existing results.

It is also important to mention how the partition recovery task can be cast as a univariate change point localization problem. Indeed, when $d = 1$, the two coincide; see [36, 35]. However, the case of $d \geq 2$ becomes significantly more challenging due to the lack of a total ordering over the lattice. Consequently, our results imply also novel localization rates for change point analysis in multivariate settings.

## 2   Consistency rates for the partition recovery problem

In this section, we investigate the theoretical properties of DCART and of a two-step estimator also based on DCART for partition recovery. We remark that instead of DCART, it is possible to deploy the ORT estimator [12] or the NP-hard estimator (2) in our algorithms. Our theoretical results still hold by simply replacing the term $k_{\mathrm{dyad}}(\theta^*)$, in both the upper bound and the choice of tuning parameters, with the smallest $k$ such that there is a $k$-hierarchical-rectangular-partition ($k_{\mathrm{hier}}(\theta^*)$) or $k$-rectangular-partition ($k(\theta^*)$) of $L_{d,n}$ associated with $\theta^*$, respectively. Thus, using these more complicated methodologies that scan over larger classes of rectangular partitions will result in smaller upper bounds in Theorems 1, 2 and 4. See [12] for details about the relationship of $k(\theta^*)$, $k_{\mathrm{dyad}}(\theta^*)$ and $k_{\mathrm{hier}}(\theta^*)$.

### 2.1   One-sided consistency: DCART

As illustrated in Figure 2, the DCART procedure will produce too fine a partition in many situations, even if the signal is directly observed (i.e., there is no noise). Thus, the naive partition estimator based on the constancy regions of the DCART estimator $\tilde{\theta}$ as in (4) will inevitably suffer from the same drawback. Nonetheless, it is still possible to demonstrate a one-sided type of accuracy and even consistency for such a simple and computationally-efficient estimator. Specifically, in our next result we show that in every dyadic rectangle supporting the DCART estimator, $\theta^*$ has almost constant mean. The reverse does not hold however, as there is no guarantee that every rectangle in the partition induced by the true signal $\theta^*$ is mostly covered by one dyadic DCART rectangle.

**Theorem 1** *Suppose that the data satisfy* (1) *and that $\tilde{\theta}$ is the DCART estimator* (4) *obtained with tuning parameter $\lambda = C\sigma^2 \log(N)$, where $C > 0$ is a sufficiently large absolute constant. Let $\{R_j\}_{j \in [k(\tilde{\theta})]}$ be the associated partition. For any $j \in [k(\tilde{\theta})]$, let $S_j \subset R_j$ be the largest subset of $R_j$ such that $\theta^*$ is constant on $S_j$. Then there exist absolute constants $C_1, C_2, C_3, C_4, C_5 > 0$ such that, with probability at least $1 - C_1 \exp\{-C_2 \log(N)\}$, the following hold:*

- *global one-sided consistency:*

$$\sum_{j \in [k(\tilde{\theta})]} |R_j \backslash S_j| \leq C_3 \kappa^{-2} \sigma^2 k_{\mathrm{dyad}}(\theta^*) \log(N); \tag{5}$$

- *local one-sided consistency: for any $j \in [k(\tilde{\theta})]$, if $R_j \setminus S_j \neq \emptyset$, then*

$$|R_j \backslash S_j| \leq C_4 \kappa_j^{-2} \sigma^2 k_{\mathrm{dyad}}(\theta^*_{R_j}) \log(N), \tag{6}$$

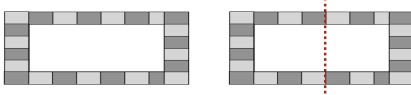

Figure 2: The left panel is the true signal. The first dyadic split always cuts a rectangle into two and ends up with over-partitioning, with the right panel as an example.

> *where $\kappa_j = \min_{s,t \in R_j : \theta_s^* \neq \theta_t^*} |\theta_s^* - \theta_t^*|$; and*

- *control on over-partitioning:*

$$k(\tilde{\theta}) \leq 2k_{\text{dyad}}(\theta^*) + C_5. \tag{7}$$

We remark that $k(\tilde{\theta}) = k_{\text{dyad}}(\tilde{\theta})$ due to the construction of $\tilde{\theta}$. Theorem 1 consists of three results. We have mentioned the over-partitioning issue of DCART. The bound (7) shows that the over-partitioning is upper bounded, in the sense that the size of the partition induced by DCART is in fact of the same order of the size of the dyadic rectangular partition associated with $\theta^*$.

For each resulting rectangle $R_j$, (6) shows that it is almost constant, in the sense that if the signal possesses different values in $R_j$, then $R_j$ includes a subset $S_j$ which has constant signal value and the size $|R_j \setminus S_j|$ is upper bounded by $\kappa_j^{-2} \sigma^2 k_{\text{dyad}}(\theta_{R_j}^*) \log(N)$, where $\kappa_j$ is the smallest jump size within $R_j$. We note that since $k_{\text{dyad}}(\theta_{R_j}^*) \leq k_{\text{dyad}}(\theta^*)$, if $k_{\text{dyad}}(\theta^*)$ is assumed to be a constant as in the cluster detection literature [e.g. 4], then for general $d \in \mathbb{N}^*$, (6) has the same estimation error rate as that in the change point detection literature [e.g. 36, 35].

The result in (6) provides an individual recovery error, with the individual jump size $\kappa_j$, while paying the price $k_{\text{dyad}}(\theta_{R_j}^*)$. In (5) we show that globally, when we add up the errors in all resulting rectangles, the overall recovery error is of order $\kappa^{-2} \sigma^2 k_{\text{dyad}}(\theta^*) \log(N)$. When $d = 1$, [35] shows the minimax rate of the $L_1$-Wasserstein distance between the vectors of true change points and of the change point estimators is of order $\kappa^{-2} \sigma^2 K$, where $K$ is the true number of change points. Comparing with this result, (5) can be seen as delivering a "one-sided" nearly optimal rate, saving for a logarithmic factor.

In the change point localization case, i.e. when $d = 1$, one can show that $k(\theta_{R_j}^*) \leq 3$ [see, e.g. 36] just assuming mild conditions on $\Delta$. However, as soon as $d \geq 2$ this is no longer the case. As an illustration, consider the left plot of Figure 2, where the whole rectangle is $R_j$ and the white one in the middle is $S_j$. Without further constraints on each component, having only conditions on $\Delta$ will not prevent a very fragmented boundary, which can increase the term $k_{\text{dyad}}(\theta_{R_j}^*)$ in (6).

## 2.2 Two-sided consistency: A two-step estimator

As we have seen in Section 2.1, despite having the penalty term $\lambda|\Pi|$ to penalize over-partitioning in the objective function (4), since the optimization of DCART only restricts to all dyadic partitions, the naive DCART estimator still suffers from over-partitioning. To address this issue, we propose a two-step estimator which builds upon DCART and merges the corresponding rectangles if they are close enough and if their estimated means are similar. The procedure will not only be guaranteed to return, with high probability, the correct number of rectangles in the partition induced by the signal $\theta^*$, but also fulfill the target for (two-sided) consistency specified in (3).

**The two-step estimator**. Our two-estimator starts with a constrained DCART, prohibiting splits resulting in rectangles that are too small. *The first step* estimation is defined as

$$\hat{\theta} = O_{S(\widehat{\Pi})}(y), \quad \text{with} \quad \widehat{\Pi} \in \operatorname*{arg\,min}_{\Pi \in \mathcal{P}_{\text{dyadic},d,n}(\eta)} \left\{ 2^{-1} \|y - O_{S(\Pi)}(y)\|^2 + \lambda_1 |\Pi| \right\}, \tag{8}$$

where $\lambda_1, \eta > 0$ are tuning parameters and $\mathcal{P}_{\text{dyadic},d,n}(\eta)$ is the set of rectangular partitions where every rectangle is of size at least $\eta$.

*The second step* merges rectangles in the partition associated with the estimator from the first step to overcome over-partitioning. To be specific, let $\{R_l\}_{l \in [k(\hat{\theta})]}$ be a rectangular partition of $L_{d,n}$

associated by $\widehat{\theta}$. For each $(i,j) \in [k(\widehat{\theta})] \times [k(\widehat{\theta})]$, $i < j$, let $z_{(i,j)} = 1$ if

$$\text{dist}(R_i, R_j) \leq \eta \tag{9}$$

and

$$\frac{1}{2}\left[\sum_{l \in R_i}(Y_l - \bar{Y}_{R_i})^2 + \sum_{l \in R_j}(Y_l - \bar{Y}_{R_j})^2\right] + \lambda_2 > \frac{1}{2}\sum_{l \in R_i \cup R_j}(Y_l - \bar{Y}_{R_i \cup R_j})^2, \tag{10}$$

where $\lambda_2 > 0$ is a tuning parameter; otherwise, we let $z_{(i,j)} = 0$. With this notation, let $E = \{(i,j) \in [k(\widehat{\theta})] \times [k(\widehat{\theta})] : z_{(i,j)} = 1\}$ and let $\{\widehat{\mathcal{C}}_l\}_{l \in [L]}$ be the collection of all the connected components of the undirected graph $([k(\widehat{\theta})], E)$. *The final output* can be written as

$$\widehat{\Lambda} = \left\{\cup_{j \in \widehat{\mathcal{C}}_1} R_j, \ldots, \cup_{j \in \widehat{\mathcal{C}}_L} R_j\right\}. \tag{11}$$

Notice that the main computational burden is to find the DCART estimator which has a cost of $O(N)$. From the output of DCART, the computation of the quantities in (9) and (10) can be done in $O(k(\widehat{\theta})^2)$. Before describing the favorable properties of the estimator, we state our main assumption.

**Assumption 1** *If $A, B \in \Lambda^*$ with $A \neq B$ and $\bar{\theta}_A^* = \bar{\theta}_B^*$, then we have that*

$$\text{dist}(A, B) \geq ck_{\text{dyad}}(\theta^*)\kappa^{-2}\sigma^2 \log(N), \tag{12}$$

*for some large enough constant $c > 0$. Furthermore, we assume that*

$$\kappa^2\Delta \geq ck_{\text{dyad}}(\theta^*)\sigma^2 \log(N). \tag{13}$$

Assumption 1 simply states that if two elements of $\Lambda^*$ have the same signal values then they should be sufficiently apart from each other. Assumption 1 also specifies a signal-to-noise ratio type of condition. When $d = 1$, it is well known in the change point literature [e.g. 36] that the optimal signal-to-noise ratio for localization is of the form $\kappa^2\Delta \gtrsim \sigma^2 \log(N)$. The condition in (13) has an additional $k_{\text{dyad}}(\theta^*)$ factor. It is an interesting open problem to determine whether this additional term can be avoided when $d \geq 2$.

**Theorem 2** *Assume that Assumption 1 holds. Suppose that the data satisfy (1) and $\widehat{\Lambda}$ is the two-step estimator, with tuning parameters $\lambda_1 = C_1\sigma^2 \log(N)$, $\lambda_2 = C_2 k_{\text{dyad}}(\theta^*)\sigma^2 \log(N)$ and*

$$c_1 k_{\text{dyad}}(\theta^*)\sigma^2\kappa^{-2} \log(N) \leq \eta \leq \Delta/c_2, \tag{14}$$

*where $c_1, c_2, C_1, C_2 > 0$ are absolute constants. Then with probability at least $1 - N^{-c}$, it holds that*

$$|\widehat{\Lambda}| = |\Lambda^*| \quad \text{and} \quad d_{\text{Haus}}(\widehat{\Lambda}, \Lambda^*) \leq C\sigma^2\kappa^{-2}k_{\text{dyad}}(\theta^*) \log(N), \tag{15}$$

*where $c, C > 0$ are absolute constants.*

Theorem 2 shows that the two-step estimator overcomes the over-partitioning issue of DCART and is consistent for the partition recovery problem provided that $\kappa^{-2}\sigma^2 k_{\text{dyad}}(\theta^*) \log(N)/\Delta \to 0$. The resulting estimation error is of order at most $\kappa^{-2}\sigma^2 k_{\text{dyad}}(\theta^*) \log(N)$.

In view of Assumption 1 and Theorem 2, intuitively speaking, (14) ensures that if there are two separated regions where the true signal takes on the same value, then the they should be far apart; otherwise, our algorithm would not be able to tell if they should be merged together or keep separated. Eq. (15) requires that the signal strength is large enough. Technically speaking, if (14) is changed to another quantity, denoted by $w$, then the final result of Theorem 2 would be

$$d_{\text{Haus}}(\widehat{\Lambda}, \Lambda^*) \leq C\left\{(w \wedge \Delta) \vee \sigma^2\kappa^{-2}k_{\text{dyad}}(\theta^*) \log(N)\right\},$$

where the term $w \wedge \Delta$ is due to the definition of $\Delta$. This means that the final rate in Theorem 2 is determined jointly by Assumption 1 and an optimal rate.

There are three tuning parameters required by the two-step estimator. Practical guidance on how to pick them will be provided in Section S6.2. The tuning parameter $\lambda_1$ in (8) is the same as the one in (4) and their theoretical rates are determined by the maximal noise level of a Gaussian distribution over all possible rectangles using a union bound argument. The tuning parameter $\lambda_2$ is used in

the merging step (10), penalizing over-partitioning. Since the candidate rectangles in (10) are the estimators from the first step, these rectangles carry the estimation error from the first step. An intermediate result from the proof of Theorem 2 unveils a similar result to Theorem 1, that for each $R_j$ involved in (10), there exists a subset $S_j \subset R_j$ having constant signal values satisfying that

$$\sum_{j \in [k(\widehat{\theta})]} |R_j \backslash S_j| \lesssim \kappa^{-2} \sigma^2 k_{\mathrm{dyad}}(\theta^*) \log(N).$$

This suggests that the right choice for $\lambda_2$ should be able to counter this extra $k_{\mathrm{dyad}}(\theta^*)$ factor. Finally, the tuning parameter $\eta$ appears twice in the estimation procedure: as a lower bound on the size of the rectangles obtained in first step as (8) and as an upper bound on the distance between two rectangles in (9); see (14). The value of $\eta$ should be at least as large as the one-sided upper bound on the recovery error, in order to ensure that over-partitioning cannot occur. As the same time, it should not be chosen too large, or otherwise the procedure may erroneously prune small true rectangles. By this logic, $\eta$ should not exceed the minimal size of the true rectangles; this is indeed the upper bound in condition (14). Finally, we would like to point out that, similar conditions are even necessary in some change point localization ($d = 1$) procedures, see [e.g. 36].

In practice, one may be tempted to abandon the tuning parameter $\eta$, and only prune the DCART output using (10). If one still wants the result to satisfy (3), then stronger conditions are needed and worse localization rates are obtained. We include this result in Section S4 in the supplementary material.

## 2.3 Optimality in the regular boundary case

We use a two-step procedure to improve the partition recovery performances of DCART and show the error is of order $\kappa^{-2} \sigma^2 k_{\mathrm{dyad}}(\theta^*) \log(N)$. A natural question in order is whether one can further expect to improve this rate. In Proposition 3, we show a minimax lower bound result.

**Proposition 3** *Let $\{y_i\}_{i \in L_{d,n}}$ satisfy (1) and*

$$\theta_i^* = \kappa, \quad \text{if } i \in S; \quad \theta_i^* = 0, \quad \text{if } i \in L_{d,n} \setminus S, \tag{16}$$

*where $S \subset L_{d,n}$ is a rectangle and $|S| = \Delta > 0$. Let $P_{\kappa,\Delta,\sigma}^N$ denote the corresponding joint distribution. Consider the class of distributions*

$$\mathcal{P}_N = \left\{ P_{\kappa,\Delta,\sigma}^N : \Delta < N/2, \ \kappa^2 \Delta / \sigma^2 \geq \log(N)/6 \right\}.$$

*Then for $N \geq 2^6$, it holds that $\inf_{\widehat{S}} \sup_{P \in \mathcal{P}_N} \mathbb{E}_P \left\{ |\widehat{S} \triangle S| \right\} \geq \sigma^2 \kappa^{-2} \log(N)/36$, where the infimum is over all estimators $\widehat{S}$ of $S$.*

Theorem 3 shows that when the induced partition of $\theta^*$ consists of one rectangle and its complement, i.e. when $k_{\mathrm{dyad}}(\theta^*) = O(1)$, the minimax lower bound on the estimation error is of order $\kappa^{-2} \sigma^2 \log(N)$. Recalling the estimation errors we derived for DCART and the two-step estimator in Theorems 1 and 2, when $k_{\mathrm{dyad}}(\theta^*) = O(1)$, the results thereof are minimax optimal.

The assumption $k_{\mathrm{dyad}}(\theta^*) = O(1)$ is fairly restrictive, though, using our notation, the case of $|\Lambda^*| = 2$ is in fact used in the cluster detection literature [4, 3]. In fact, in order to achieve the optimal estimation rate indicated in Theorem 3, we only need a boundary regularity condition, in the sense that for every rectangle in the partition induced by $\theta^*$, there are only $O(1)$-many other rectangles nearby. This condition is formally stated in Assumption 2.

**Assumption 2** *There exists constant $C, c > 0$ such that for any $A \in \Lambda^*$ it holds that*

$$\left| \left\{ B \in \Lambda^* \backslash \{A\} : \quad \mathrm{dist}(A, B) \leq c\sigma^2 \kappa^{-2} k_{\mathrm{dyad}}(\theta^*) \log(N) \right\} \right| \leq C.$$

Assumption 2 asserts that within $c\kappa^{-2}\sigma^2 k_{\mathrm{dyad}}(\theta^*) \log(N)$ distance, each element of $\Lambda^*$ only has $O(1)$-many elements nearby. This condition shares the same spirit of requiring the cluster boundary to be a bi-Lipschitz function in the cluster detection literature [e.g. 2].

Table 1: Performance evaluations over 50 repetitions under different scenarios. The performance metrics $\text{dist}_1$ and $\text{dist}_2$ are defined in the text. The numbers in parenthesis denote standard errors.

| Setting | | dist$_1$ | | dist$_2$ | | Setting | | dist$_1$ | | dist$_2$ | |
|---|---|---|---|---|---|---|---|---|---|---|---|
| | $\sigma$ | $\widehat{\Lambda}$ | TV-based | $\widehat{\Lambda}$ | TV-based | | $\sigma$ | $\widehat{\Lambda}$ | TV-based | $\widehat{\Lambda}$ | TV-based |
| 1 | 0.5 | **35.8(12.2)** | 51.6(21.9) | **0.0(0.0)** | 0.1(0.3) | 2 | 0.5 | 462.6(637.7) | **418.6(246.9)** | **0.2(0.4)** | 0.4(0.6) |
| 1 | 1.0 | **196.1(401.8)** | 582.3(2429.5) | **0.0(0.2)** | 0.3(0.5) | 2 | 1.0 | **2617.7(4047.7)** | 8630.6(6049.3) | **0.7(0.6)** | 1.4(0.7) |
| 1 | 1.5 | **298.0(878.7)** | 4513.6(5970.9) | **0.1(0.3)** | 0.5(0.5) | 2 | 1.5 | **4706.7(5213.7)** | 11477.3(5213.6) | **1.2(0.9)** | 1.9(0.4) |
| 3 | 0.5 | **62.1(231.4)** | 123.0(44.1) | **0.0(0.1)** | 0.2(0.4) | 4 | 0.5 | 86.3( 231.1) | **52.8(21.7)** | **0.2(0.4)** | 0.3(0.4) |
| 3 | 1.0 | **150.8(227.3)** | 1012.7(752.5) | **0.1(0.3)** | 0.7(0.5) | 4 | 1.0 | 119.6(189.3) | **87.8(82.0)** | **0.2(0.4)** | 1.1(1.1) |
| 3 | 1.5 | **270.8(479.0)** | 12732.6(3296.8) | **0.2(0.5)** | 1.9(0.4) | 4 | 1.5 | 399.3(437.1) | **217.6(233.5)** | **0.4(0.7)** | 1.4(1.1) |

**Corollary 4** *Assume that Assumptions 1 and 2 hold. Suppose that the data satisfy* (1) *and* $\widehat{\Lambda}$ *is the two-step estimator defined in* (11), *with tuning parameters* $\lambda_1 = C_1 \sigma^2 \log(N)$, $\lambda_2 = C_2 k_{\text{dyad}}(\theta^*) \sigma^2 \log(N)$ *and* $c_1 \kappa^{-2} k_{\text{dyad}}(\theta^*) \sigma^2 \log(N) \leq \eta \leq \Delta/c_2$, *where* $c_1, c_2, C_1, C_2 > 0$ *are absolute constants. Then with probability at least* $1 - N^{-c}$, *it holds that* $|\widehat{\Lambda}| = |\Lambda^*|$ *and*

$$\max_{A \in \Lambda^*} \min_{\widehat{A} \in \widehat{\Lambda}} |\widehat{A} \triangle A^*| \leq \frac{C\sigma^2 \log(N)}{\kappa^2} \min \left\{ k(\theta^*), \max_{B \in \Lambda^*, \, \text{dist}(A,B) \leq c\sigma^2 \kappa^{-2} k_{\text{dyad}}(\theta^*) \log(N)} |B|/\eta \right\}.$$

(17)

*where* $c, C > 0$ *are absolute constants. In particular, if* $\eta \asymp \Delta$ *and* $|A| \asymp \Delta$, *for all* $A \in \Lambda^*$, *then*

$$d_{\text{Haus}}(\widehat{\Lambda}, \Lambda^*) \leq C \sigma^2 \kappa^{-2} \log(N).$$

(18)

Corollary 4 shows that even if $k_{\text{dyad}}(\theta^*)$ is diverging as the sample size grows unbounded, one can still achieve the minimax optimal estimation error rate $\kappa^{-2}\sigma^2 \log(N)$, with properly chosen tuning parameters and additional regularity conditions on the partition. An interesting by-product in deriving this rate is (17), which characterizes the effect of the number of nearby rectangles in the estimation error for individual elements in $\Lambda^*$.

## 3 Experiments

In this section, we demonstrate in simulation the numerical performances of the two-step estimator for the task of partition recovery. The code is by courtesy of the authors of [12] and all of our experiments are done in a 2.3 GHz 8-Core Intel Core i9 machine. Our code can be found in `https://github.com/hernanmp/Partition_recovery`. We focus on the naive two-step estimator detailed in Section S4 and denoted here as $\widehat{\Lambda}$. The implementation details regarding choice of tuning parameters are discussed in S6.2.

We adopt $\text{dist}_1 = d_{\text{Haus}}(\widehat{\Lambda}, \Lambda^*)$ and $\text{dist}_2 = ||\widehat{\Lambda}| - |\Lambda^*||$ as the measurements. For each scenario depicted in Figure 3, we report the mean and standard errors of $\text{dist}_1$ and $\text{dist}_2$ over 50 Monte Carlo simulations.

As a competitor benchmark, we consider a similar pruning algorithm based on the total variation estimator [27, 33], namely TV-based, instead of ours based on DCART. The implementation details are discussed in Section S6.3.

For each scenario considered, we vary the noise level as $\sigma \in \{0.5, 1, 1.5\}$ and set $(d, n) = (2, 2^7)$. In each instance, the data are generated as $y \sim \mathcal{N}(\theta^*, \sigma^2 I_{L_{d,n}})$. Detailed descriptions are in Section S6.1, and visualizations of the signal patterns are in the second column in Figure 3, while the third and fourth columns depict $\tilde{\theta}$, the DCART estimator, and $\widehat{\theta}$, our two-step estimator, respectively. We can see that our two-step estimator correctly identifies the partition and improves upon DCART for the purpose of recovery partition. It is worth noting that even when the partition is not rectangular, as shown in the second row in Figure 3, our two-step estimator is still able to accurately recover a good rectangular partition.

From Table 1 we see that in terms of the metric $\text{dist}_2$ our two-step estimator outperforms TV-based estimator in all cases. Furthermore, the same is also true for most cases in terms of the metric $\text{dist}_1$.

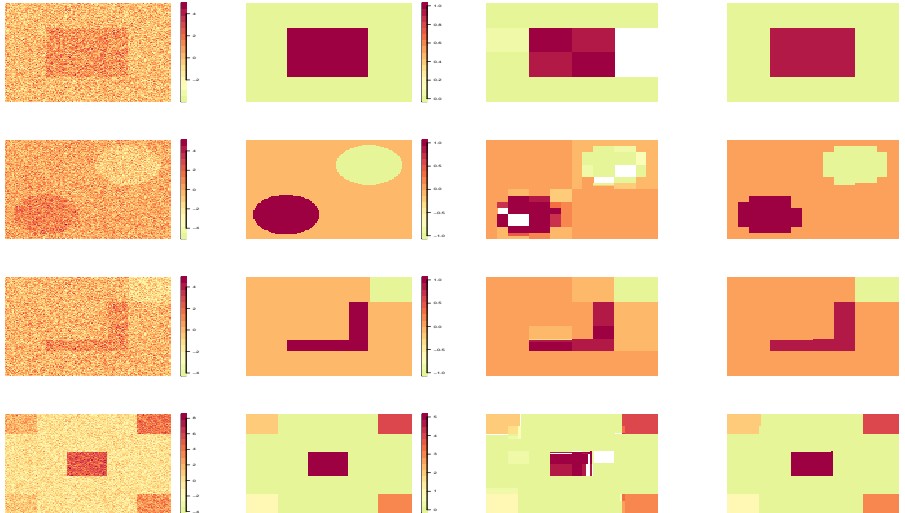

Figure 3: From top to bottom: Scenarios 1 to 4. From left to right: An instance of $y$, the signal $\theta^*$, DCART, and DCART after merging. In each case the data are generated with $\sigma = 1$.

# 4   Conclusions

In this paper we study the partition recovery problem over $d$-dimensional lattices. We show how a simple modification of DCART enjoys one-sided consistency. To correctly identify the size of the true partition and obtain better consistency guarantees, we further propose a more sophisticated two-step estimator which is shown to be minimax optimal in some regular cases.

Throughout this paper, we discuss partition recovery on lattice grids. In fact, to deal with non axis-aligned data, one can construct a lattice in the domain of the features, average the observations within each cell of the lattice and ignore the cells without observations. More details can be found in the supplementary materials.

Finally, an important open question remains regarding the necessity to include the factor $k_{\mathrm{dyad}}(\theta^*)$ in the signal-to-noise condition (13) and in the estimation rates. We leave for future work to investigate what the optimal estimation rates are in general when $k_{\mathrm{dyad}}(\theta^*)$ is allowed to diverge.

## Supplementary material

The supplementary material contains all the technical details, additional simulation results and code of this paper.

## Acknowledgments and Disclosure of Funding

Funding in direct support of this work: NSF DMS 2015489 and EPSRC EP/V013432/1.

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
