# Supplementary Material for "Lattice partition recovery with dyadic CART"

**Oscar Hernan Madrid Padilla**
Department of Statistics
University California, Los Angeles
oscar.madrid@stat.ucla.edu

**Yi Yu**
Department of Statistics
University of Warwick
yi.yu.2@warwick.ac.uk

**Alessandro Rinaldo**
Department of Statistics & Data Science
Carnegie Mellon University
arinaldo@cmu.edu

In this document, we provide some further technical details and all the proofs of the results in "Lattice partition recovery with dyadic CART".

## S1 Comparisons with some existing results

The procedures proposed in this paper all rely crucially on the DCART estimator $\tilde{\theta}$ defined in (4). As shown recently in [4], $\tilde{\theta}$ is such that $\mathbb{E}\{\|\tilde{\theta} - \theta^*\|^2\} \lesssim \sigma^2 k_{\mathrm{dyad}}(\theta^*) \log(N)$, a rate that is the minimax optimal. [4] also studies the de-noising performances of other rectangular partition estimators. [5] studied the de-noising performances of an $\ell_0$-penalized estimator for a structured signal supported over general graphs and obtained the same rates. Both the DCART and the estimator proposed in [5] are based on $\ell_0$-penalization. A different approach is to instead rely on $\ell_1$-penalizations [e.g. 9, 7].

In light of the de-noising rate, it is perhaps not surprising that the partition recovery estimation error rate of the DCART, shown in Theorem 1 is of order "de-noising error bound/jump size", but what is unsatisfactory for us is that when $d = 1$, this extra $k_{\mathrm{dyad}}(\theta^*)$ factor suggests the sub-optimality of the result. For instance, both [12] and [11] showed that when $d = 1$, an $\ell_0$-penalized estimator is able to achieve a minimax optimal estimation error of order $\kappa^{-2}\sigma^2 \log(N)$. In Section 2.3, we have shown that the term $k_{\mathrm{dyad}}(\theta^*)$ can be avoided if further regularity condition is imposed. It remains still an open problem without these regularity conditions, what the optimal estimation rate would be.

It is also worth mentioning another stream of work, focusing on the detection boundary in detection a cluster of nodes in general graphs, including square lattices. Although testing and estimation are two fundamentally different problems, often requiring different conditions, the detection boundaries derived thereof could be a useful reference evaluating the signal-to-noise ratio condition we impose in (9). [3, 2, 1], among others, stated that the detection boundary, in our notation is $\kappa^2\Delta \asymp$ a logarithmic term. Such rate is derived for $k(\theta^*) = O(1)$ and suggests that our condition (9) is optimal when $k(\theta^*) = O(1)$. It remains an open problem to determine the optimal estimation rate when $k(\theta^*)$ is allowed to diverge.

## S2 Additional definitions

We have repeatedly used a concept that two rectangles are adjacent. In addition to the explanation in Definition 2, we detail all the possible situations in Definition S1 below.

**Definition S1.** *For two disjoint subsets $R_1, R_2 \subset L_{d,n}$, with $d > 1$, $R_l = \prod_{i=1}^{d}[a_i^{(l)}, b_i^{(l)}]$, $l \in \{1, 2\}$, we say that $R_1$ and $R_2$ are adjacent if there exists $i_0 \in [d]$, such that one of the following holds:*

35th Conference on Neural Information Processing Systems (NeurIPS 2021).

- $b_{i_0}^{(1)} - a_{i_0}^{(2)} = 1$ *and* $\prod_{i \neq i_0}[a_i^{(1)}, b_i^{(1)}] \subset \prod_{i \neq i_0}[a_i^{(2)}, b_i^{(2)}]$;

- $b_{i_0}^{(1)} - a_{i_0}^{(2)} = 1$ *and* $\prod_{i \neq i_0}[a_i^{(2)}, b_i^{(2)}] \subset \prod_{i \neq i_0}[a_i^{(1)}, b_i^{(1)}]$;

- $a_{i_0}^{(1)} - b_{i_0}^{(2)} = 1$ *and* $\prod_{i \neq i_0}[a_i^{(1)}, b_i^{(1)}] \subset \prod_{i \neq i_0}[a_i^{(2)}, b_i^{(2)}]$;

- $a_{i_0}^{(1)} - b_{i_0}^{(2)} = 1$ *and* $\prod_{i \neq i_0}[a_i^{(2)}, b_i^{(2)}] \subset \prod_{i \neq i_0}[a_i^{(1)}, b_i^{(1)}]$.

## S3  Proofs of main results

This section contains the proofs of the main results from Section 2. Theorem 1 demonstrates the one-sided consistency of DCART. This result is not only interesting on its own but is also used repeatedly and in as essential away to prove two-sided consistency. For readability, we express the two main claims of Theorem 1, namely (5) and (6), as the events

$$\mathcal{A}_1 = \left\{ \sum_{j \in [k(\tilde{\theta})]} |R_j \backslash S_j| \leq C_3 \kappa^{-2} \sigma^2 k_{\mathrm{dyad}}(\theta^*) \log(N) \right\} \tag{S1}$$

and

$$\mathcal{A}_2 = \left\{ |R_j \backslash S_j| \leq C_4 \kappa_j^{-2} \sigma^2 k_{\mathrm{dyad}}(\theta_{R_j}^*) \log(N), \quad j \in [k(\tilde{\theta})] \text{ and } R_j \setminus S_j \neq \emptyset \right\}, \tag{S2}$$

respectively

### S3.1  One-sided consistency of DCART

*Proof of Theorem 1.* For $j \in [k(\tilde{\theta})]$, if $R_j \setminus S_j \neq \emptyset$, then let $r_j$ be the smallest positive integer such that there exists a partition of $R_j$, namely $\{T_{j,1}, \ldots, T_{j,r_j}, S_j\}$ with $\theta_i^* = a_{j,l}$, for all $i \in T_{j,l}$, $l \in [r_j]$.

Without loss of generality assume that $0 < |T_{j,1}| \leq |T_{j,2}| \leq \ldots \leq |T_{j,r_j}| \leq |S_j|$, for each $j \in [k(\tilde{\theta})]$. Suppose that $r_j$ is even. Then

$$
\begin{aligned}
|R_j \backslash S_j| &= \sum_{l=1}^{r_j/2} |T_{j,2l-1}| + \sum_{l=1}^{r_j/2} |T_{j,2l}| \\
&= \sum_{l=1}^{r_j/2} \min\{|T_{j,2l-1}|, |T_{j,2l}|\} + \sum_{l=1}^{r_j/2-1} \min\{|T_{j,2l}|, |T_{j,2l+1}|\} + \min\{|T_{r_j}|, |S_j|\} \\
&\leq 2 \sum_{l=1}^{r_j/2} \frac{|T_{j,2l-1}| \, |T_{j,2l}|}{|T_{j,2l-1}| + |T_{j,2l}|} + 2 \sum_{l=1}^{r_j/2-1} \frac{|T_{j,2l}| \, |T_{j,2l+1}|}{|T_{j,2l}| + |T_{j,2l+1}|} + 2 \frac{|T_{r_j}| \, |S_j|}{|T_{r_j}| + |S_j|} \\
&\leq \frac{2}{\kappa_j^2} \sum_{l=1}^{r_j/2} \frac{|T_{j,2l-1}| \, |T_{j,2l}|}{|T_{j,2l-1}| + |T_{j,2l}|} (a_{j,2l-1} - a_{j,2l})^2 \\
&\quad + \frac{2}{\kappa_j^2} \sum_{l=1}^{r_j/2-1} \frac{|T_{j,2l}| \, |T_{j,2l+1}|}{|T_{j,2l}| + |T_{j,2l+1}|} (a_{j,2l} - a_{j,2l+1})^2 + \frac{2}{\kappa_j^2} \frac{|S_j| \, |T_{r_j}|}{|S_j| + |T_{r_j}|} (\bar{\theta}_{S_j}^* - a_{r_j})^2 \\
&\leq \frac{2}{\kappa_j^2} \sum_{l=1}^{r_j/2} \sum_{i \in T_{j,2l-1} \cup T_{j,2l}} (\theta_i^* - \bar{\theta}_{T_{j,2l-1} \cup T_{j,2l}}^*)^2 \\
&\quad + \frac{2}{\kappa_j^2} \sum_{l=1}^{r_j/2-1} \sum_{i \in T_{j,2l} \cup T_{j,2l+1}} (\theta_i^* - \bar{\theta}_{T_{j,2l} \cup T_{j,2l+1}}^*)^2 + \frac{2}{\kappa_j^2} \sum_{i \in S_j \cup T_{r_j}} (\theta_i^* - \bar{\theta}_{S_j \cup T_{r_j}}^*)^2 \\
&\leq \frac{2}{\kappa_j^2} \sum_{l=1}^{r_j/2} \sum_{i \in T_{j,2l-1} \cup T_{j,2l}} (\theta_i^* - \bar{\theta}_{R_j}^*)^2 + \frac{2}{\kappa_j^2} \sum_{l=1}^{r_j/2-1} \sum_{i \in T_{j,2l} \cup T_{j,2l+1}} (\theta_i^* - \bar{\theta}_{R_j}^*)^2 \\
&\quad + \frac{2}{\kappa_j^2} \sum_{i \in S_j \cup T_{r_j}} (\theta_i^* - \bar{\theta}_{R_j}^*)^2 \leq \frac{4}{\kappa_j^2} \sum_{i \in R_j} (\theta_i^* - \bar{\theta}_{R_j}^*)^2,
\end{aligned}
$$

where the first inequality follows from Lemma S2. The same bounds holds also when $r_j$ is odd. Hence,

$$
\begin{aligned}
|R_j \backslash S_j| &\leq \frac{8}{\kappa_j^2} \sum_{i \in R_j} (\theta_i^* - \bar{y}_{R_j})^2 + \frac{8}{\kappa_j^2} \sum_{i \in R_j} (\bar{\theta}_{R_j}^* - \bar{y}_{R_j})^2 \\
&= \frac{8}{\kappa_j^2} \sum_{i \in R_j} (\theta_i^* - \tilde{\theta}_i)^2 + \frac{8}{\kappa_j^2} |R_j| (\bar{\theta}_{R_j}^* - \bar{y}_{R_j})^2.
\end{aligned}
\tag{S3}
$$

Let $\Omega_1$ and $\Omega_3$ be the events defined below in (S18) and (S24), respectively. In the event $\Omega_1 \cap \Omega_3$, the result (6), i.e. the event $\mathcal{A}_2$ defined in (S2), is a direct consequence of (S3).

Let $\Omega_2$ be the event defined in (S22). In the event $\Omega_1 \cap \Omega_2 \cap \Omega_3$, it follows from (S3) that (5), i.e. the event $\mathcal{A}_1$ defined in (S1), holds. To be specific, we have that

$$
\begin{aligned}
\sum_{j=1}^{k(\tilde{\theta})} |R_j \backslash S_j| &\leq \sum_{j=1}^{k(\tilde{\theta})} \left[ \frac{8}{\kappa^2} \sum_{i \in R_j} (\theta_i^* - \tilde{\theta}_i)^2 + \frac{8}{\kappa^2} |R_j| (\bar{\theta}_{R_j}^* - \bar{y}_{R_j})^2 \right] \\
&\leq \frac{8}{\kappa^2} \|\theta^* - \tilde{\theta}\|^2 + \frac{8 c_1 c_3 \sigma^2 k_{\mathrm{dyad}}(\theta^*) \log(N)}{\kappa^2} \leq \frac{C_3 \sigma^2 \log(N) k_{\mathrm{dyad}}(\theta^*)}{\kappa^2}.
\end{aligned}
$$

Finally, note that the final theorem claim (7) is shown in Lemma S5. $\qquad\square$

## S3.2 Two-sided consistency of DCART: a two-step constrained estimator

*Proof of Theorem 2.* The proof of (15) is identical to that of Theorem S1 with one difference. The rectangular partition induced by $\widehat{\theta}$ is such that

$$
\min\{|R_i|, |R_j|\} \geq \eta \geq c_1 \frac{k_{\mathrm{dyad}}(\theta^*) \sigma^2 \log(N)}{\kappa^2}.
$$

As a result, we do not need to account for the term $\sum_{j \,:\, |R_j| \leq \eta} |R_j|$, and this is the only part of the proof of Theorem S1 that requires the stronger requirement in Assumption S1. The rest of the proof goes through using Assumption 1. $\qquad\square$

## S3.3 Optimality: A regular boundary case

*Proof of Corollary 4.* Let $\widehat{\theta}$ be the estimator of $\theta^*$ defined in (10). Let $\{R_l\}_{l \in [k(\widehat{\theta})]}$ be a rectangular partition of $L_{d,n}$ induced by $\widehat{\theta}$ and let $S_j$ be the largest subset of $R_j$ with constant $\theta^*$ value, for $j \in [k(\widehat{\theta})]$. Let $\mathcal{J} \subset [k(\widehat{\theta})]$, such that $R_j \backslash S_j \neq \emptyset$, $j \in \mathcal{J}$. With the notation in Theorem 1, define the event $\mathcal{A}_3$ as

$$
\mathcal{A}_3 = \left\{ |R_j \backslash S_j| \leq C_4 \frac{\sigma^2 k_{\mathrm{dyad}}(\theta_{R_j}^*) \log(N)}{\kappa^2}, \quad j \in \mathcal{J} \right\} \cap \left\{ k(\widehat{\theta}) \leq c_1 k_{\mathrm{dyad}}(\theta^*) \right\}.
\tag{S4}
$$

It follows from (6) that the event $\mathcal{A}_3$ holds with probability at least $1 - N^{-c}$ for some positive constants $c, C_4$ and $c_1$. The rest of this proof is conducted in the event $\mathcal{A}_3$.

For any $j \in \mathcal{J}$. Let $A, B \in \Lambda^*$ be that $A \neq B$, $\bar{\theta}_A^* = \bar{\theta}_B^*$ and $(R_j \cap A) \cup (R_j \cap B) \subset S_j$. Then it follows from an almost identical argument as that in **Step 1.1** in the proof of Theorem S1, we see that Assumption 1 leads to a contradiction. It follows that $S_j$ is a connected set in $L_{d,n}$. Hence, we let $A \in \Lambda^*$ be such that $R_j \cap A = S_j$.

Suppose now that $B \in \Lambda^*$ and $R_j \cap B \subset R_j \backslash S_j$. Since

$$
|R_j \backslash S_j| \leq C_4 \frac{\sigma^2 k_{\mathrm{dyad}}(\theta_{R_j}^*) \log(N)}{\kappa^2} \leq C_4 \frac{\sigma^2 k_{\mathrm{dyad}}(\theta^*) \log(N)}{\kappa^2},
$$

recalling that $\mathrm{dist}(A, B) = \min_{a \in A, b \in B} \|a - b\|$, it holds that

$$
\mathrm{dist}(A, B) \leq C_5 \frac{\sigma^2 k_{\mathrm{dyad}}(\theta^*) \log(N)}{\kappa^2}
$$

for some constant $C_5 > 0$. Hence,

$$|\{B \in \Lambda^* \,:\, R_j \cap B \subset R_j \backslash S_j\}| \leq \left|\left\{B \in \Lambda^* \backslash \{A\} : \mathrm{dist}(A,B) \leq \frac{c\sigma^2 k_{\mathrm{dyad}}(\theta^*)\log(N)}{\kappa^2}\right\}\right| \leq C,$$

where the second inequality follows from Assumption 2. As a result,

$$|R_j \backslash S_j| \; \leq \; C_4 C \frac{\sigma^2 \log(N)}{\kappa^2}. \tag{S5}$$

It follows from an identical argument as that in **Step 5** in the proof of Theorem S1 that, for any $A \in \Lambda^*$ there exists $\hat{A} \in \widehat{\Lambda}$ such that

$$|\hat{A} \backslash A| \leq \sum_{j \in I_A} |R_j \backslash S_j| \leq C_4 C \frac{\sigma^2 \log(N)}{\kappa^2} |\{j \,:\, R_j \cap A = S_j\}|$$

$$\leq C_4 C \frac{\sigma^2 k(\widehat{\theta})\log(N)}{\kappa^2} \leq C_6 \frac{\sigma^2 k_{\mathrm{dyad}}(\theta^*)\log(N)}{\kappa^2}, \tag{S6}$$

for some constant $C_6 > 0$, where the second inequality follows from (S5). Hence,

$$|\{j \,:\, R_j \cap A = S_j\}| \leq \eta^{-1}\sum_{j \in I_A} |R_j| \leq |\hat{A}|/\eta \leq (|A| + |\hat{A} \backslash A|)/\eta$$

$$\leq |A|/\eta + C_6 \frac{\sigma^2 k_{\mathrm{dyad}}(\theta^*)\log(N)}{\eta\kappa^2} \leq \frac{C'|A|}{\eta},$$

where $C' > 0$ is an absolute constant. Combining the above with (S6) we arrive at

$$|\hat{A} \backslash A| \leq \frac{C''\sigma^2 \log(N)}{\kappa^2}\frac{|A|}{\eta}.$$

To bound the difference from the other direction, we have that

$$|A \backslash \hat{A}| \leq \sum_{j \,:\, R_j \cap A \notin \{\emptyset, S_j\}} |R_j \backslash S_j|$$

$$\leq \sum_{j \,:\, \exists B \in \Lambda^*, \; R_j \cap B = S_j, \; \mathrm{dist}(A,B) \leq c\sigma^2\kappa^{-2}k_{\mathrm{dyad}}(\theta^*)\log(N)} |R_j \backslash S_j|$$

$$\leq C_7 \frac{\sigma^2 \log(N)}{\kappa^2} \left|\{j \,:\, \exists B \in \Lambda^*, \; R_j \cap B = S_j, \; \mathrm{dist}(A,B) \leq c\sigma^2\kappa^{-2}k_{\mathrm{dyad}}(\theta^*)\log(N)\}\right|$$

$$\leq C_7 \frac{\sigma^2 \log(N)}{\kappa^2} k(\widehat{\theta}) \leq C_8 \frac{\sigma^2 \log(N)}{\kappa^2} k_{\mathrm{dyad}}(\theta^*), \tag{S7}$$

for some constants $C_7, C_8 > 0$, where the second inequality follows from Assumption 1, and the last from (S5). Hence,

$$\left|\left\{j \,:\, \exists B \in \Lambda^*, \; R_j \cap B = S_j, \; \mathrm{dist}(A,B) \leq \frac{c\sigma^2 k_{\mathrm{dyad}}(\theta^*)\log(N)}{\kappa^3}\right\}\right|$$

$$\leq \frac{1}{\eta}\sum_{B \in \Lambda^*, \; \mathrm{dist}(A,B) \leq c\sigma^2\kappa^{-2}k_{\mathrm{dyad}}(\theta^*)\log(N)} \; \sum_{j \in I_B} |R_j|$$

$$= \frac{1}{\eta}\sum_{B \in \Lambda^*, \; \mathrm{dist}(A,B) \leq c\sigma^2\kappa^{-2}k_{\mathrm{dyad}}(\theta^*)\log(N)} |\hat{B}|$$

$$\leq \frac{1}{\eta}\sum_{B \in \Lambda^*, \; \mathrm{dist}(A,B) \leq c\sigma^2\kappa^{-2}k_{\mathrm{dyad}}(\theta^*)\log(N)} \left[|B| + |B \backslash \hat{B}|\right]$$

$$\lesssim \max_{B \in \Lambda^*, \; \mathrm{dist}(A,B) \leq c\sigma^2\kappa^{-2}k_{\mathrm{dyad}}(\theta^*)\log(N)} \frac{|B|}{\eta} + \max_{B \in \Lambda^*, \; \mathrm{dist}(A,B) \leq c\sigma^2\kappa^{-2}k_{\mathrm{dyad}}(\theta^*)\log(N)} \frac{|B \backslash \hat{B}|}{\eta}$$

$$\lesssim \max_{B \in \Lambda^*, \; \mathrm{dist}(A,B) \leq c\sigma^2\kappa^{-2}k_{\mathrm{dyad}}(\theta^*)\log(N)} \frac{|B|}{\eta} + \frac{\sigma^2 \log(N)}{\eta\kappa^2} k_{\mathrm{dyad}}(\theta^*)$$

$$\lesssim \max_{B \in \Lambda^*, \; \mathrm{dist}(A,B) \leq c\sigma^2\kappa^{-2}k_{\mathrm{dyad}}(\theta^*)\log(N)} \frac{|B|}{\eta},$$

where the third inequality follows from Assumption 2, the fourth from (S7), and the last from (14). We therefore have shown (17). The claim (18) is a straightforward consequence of (17) by letting $\eta \asymp |A| \asymp \Delta$. $\qquad\square$

*Proof of Proposition 3.* We are using Fano's method in this proof. To be specific, we are to use the version of Lemma 3 in [13].

Without loss of generality, we assume that $\Delta^{1/d}$ is a positive integer. For $q$ to be specified, we further assume that $(n - \Delta^{1/d})/q$ and $q$ are both positive integers. We construct a collection of distributions, each of which is defined uniquely with a subset $S$ defined in (16). Therefore the collection of distributions can be specified by the collection of subsets

$$\mathcal{S} = \left\{ \prod_{p=1}^{d} [k_p q, k_p q + \Delta^{1/d}], \quad (k_1, \ldots, k_d) \in [0, (n - \Delta^{1/d})/q]^d \right\}.$$

We assume that the parameters $\kappa, \sigma, \Delta$ in this collection of distributions ensure that this collection of distributions belong to the subset $\mathcal{P} \subset \mathcal{P}_N$,

$$\mathcal{P} = \left\{ P_{\kappa, \Delta, \sigma}^N : \Delta^{2d} \leq N, \ \kappa^2 \Delta / \sigma^2 = \log(N)/6 \right\}. \tag{S8}$$

To justify the conditions of Lemma 3 in [13], we first notice that for each $S \in \mathcal{S}$, $|S| = \Delta$. Secondly, for any $S_1, S_2 \in \mathcal{S}$, $S_1 \neq S_2$, it holds that

$$|S_1 \triangle S_2| \geq 2\Delta^{\frac{d-1}{d}} q$$

and

$$\mathrm{KL}(P_{S_1}, P_{S_2}) \leq \Delta \kappa^2 / \sigma^2.$$

Lastly, we note that $|\mathcal{S}| = (n - \Delta^{1/d})^d / q^d$. Then Lemma 3 in [13] shows that

$$\inf_{\widehat{S}} \sup_{P \in \mathcal{P}_N} \mathbb{E}_P \left\{ |\widehat{S} \triangle S| \right\} \geq \inf_{\widehat{S}} \sup_{P \in \mathcal{P}} \mathbb{E}_P \left\{ |\widehat{S} \triangle S| \right\} \geq \Delta^{\frac{d-1}{d}} q \left( 1 - \frac{\Delta \kappa^2 / \sigma^2 + \log(2)}{\log \left\{ (n - \Delta^{1/d})^d / q^d \right\}} \right).$$

We now take $q = \Delta^{1/d}/2$, such that due to the conditions in (S8), it holds that

$$\Delta^{\frac{d-1}{d}} q = \Delta/2 = \frac{\sigma^2 \log(N)}{12 \kappa^2}$$

and have that

$$\inf_{\widehat{S}} \sup_{P \in \mathcal{P}_N} \mathbb{E}_P \left\{ |\widehat{S} \triangle S| \right\} \geq \frac{\sigma^2 \log(N)}{12 \kappa^2} \left( 1 - \frac{\log(N)/3}{\log(N)/2} \right) \geq \frac{d \sigma^2 \log(n)}{36 \kappa^2},$$

where the first inequality holds provided $6 \log(2) \leq d \log(n)$ and the conditions specified in (S8). $\qquad \square$

## S4  A naive two-step estimator

In Section 2.2, we proposed and studied a two-step constrained estimator, which builds and improve upon the DCART estimator, leading to a two-sided consistency guarantee for recovering the support of the true partition. The two-step estimator studied in Section 2.2 starts with a constrained DCART estimator and prunes its output by merging certain pairs of rectangles. It is natural to ask about the performances of a naive two-step estimator, which just prunes the DCART estimator without constraining it to only output large enough rectangles. In this section, we study thee performance of this simpler estimator, which turns out to be worse than the two-step estimator studied in Section 2.2. The proof of Theorem S1 is repeatedly used in the proofs of two of our main results, Theorem 2 and Corollary 4.

Instead of requiring Assumption 1 as in Section 2.2, we impose a stronger assumption below.

**Assumption S1.** *If $A, B \in \Lambda^*$ with $A \neq B$ and $\bar{\theta}_A^* = \bar{\theta}_B^*$, then we have that*

$$\mathrm{dist}(A, B) \geq c \frac{k_{\mathrm{dyad}}(\theta^*)^2 \sigma^2 \log(N)}{\kappa^2},$$

*for some large enough constant $c > 0$. Furthermore, we assume that*

$$\kappa^2 \Delta \geq c k_{\mathrm{dyad}}(\theta^*)^2 \sigma^2 \log(N). \tag{S9}$$

We first detail the pruning step of the naive two-step estimator. Let $\tilde{\theta}$ be the DCART estimator with tuning parameter $\lambda_1$, defined in (4). Let $\{R_l\}_{l \in [k(\tilde{\theta})]}$ be a rectangular-partition of $L_{d,n}$ induced by $\tilde{\theta}$. Let $\lambda_2, \eta, \gamma > 0$ be tuning parameters for the pruning stage. For each $(i,j) \in [k(\tilde{\theta})] \times [k(\tilde{\theta})]$, let $z_{(i,j)} = 1$ if

$$\mathrm{dist}(R_i, R_j) \le \gamma, \quad \min\{|R_i|, |R_j|\} \ge \eta$$

and

$$\frac{1}{2}\left[\sum_{l \in R_i}(Y_l - \bar{Y}_{R_i})^2 + \sum_{l \in R_j}(Y_l - \bar{Y}_{R_j})^2\right] + \lambda_2 > \frac{1}{2}\sum_{l \in R_i \cup R_j}(Y_l - \bar{Y}_{R_i \cup R_j})^2;$$

otherwise, let $z_{(i,)} = 0$. With this notation, let $E = \{(i,j) \in [k(\tilde{\theta})] \times [k(\tilde{\theta})] : z_{(i,j)} = 1\}$ and let $\{\mathcal{C}_l\}_{l \in [\hat{L}]}$ be the collection of all the connected components of the undirected graph $G_{\mathrm{naive}} := ([k(\tilde{\theta})]\backslash\mathcal{I}, E)$, where $\mathcal{I} = \{i \in [k(\tilde{\theta})] : |R_i| \le \eta\}$. Then assign each element $i \in \mathcal{I}$ at random to one of the components $\{\mathcal{C}_l\}_{l \in [\check{L}]}$ and denote the resulting collection as $\{\check{\mathcal{C}}_l\}_{l \in [\check{L}]}$. Finally, define

$$\check{\Lambda} = \left\{\cup_{j \in \check{\mathcal{C}}_1} R_j, \ldots, \cup_{j \in \check{\mathcal{C}}_{\check{L}}} R_j\right\}. \tag{S10}$$

**Theorem S1.** *Suppose Assumption S1 holds and that the data satisfy* (1) *and let $\check{\Lambda}$ be the naive two-step estimator defined in* (S10), *with tuning parameters* $\lambda_1 = C_1\sigma^2 \log(N)$, $\lambda_2 = C_2 k_{\mathrm{dyad}}(\theta^*)\sigma^2 \log(N)$, $\gamma = C_\gamma k_{\mathrm{dyad}}(\theta^*)\eta$ *and*

$$c_1 \frac{k_{\mathrm{dyad}}(\theta^*)\sigma^2 \log(N)}{\kappa^2} \le \eta \le \frac{\Delta}{c_2 k_{\mathrm{dyad}}(\theta^*)}, \tag{S11}$$

*where $C_1, C_2, C_\gamma, c_1, c_2 > 0$ are absolute constants. Then, with probability at least $1 - N^{-c}$, it holds that*

$$|\check{\Lambda}| = |\Lambda^*| \quad and \quad d_{\mathrm{Haus}}(\check{\Lambda}, \Lambda^*) \le C k_{\mathrm{dyad}}(\theta^*)\eta,$$

*where $c, C > 0$ are absolute constants.*

*If in addition, it holds that $\eta = C_\eta \kappa^{-2} k_{\mathrm{dyad}}(\theta^*)\sigma^2 \log(N)$, where $C_\eta > 0$ is an absolute constant, then, with proability at least $1 - N^{-c}$,*

$$|\check{\Lambda}| = |\Lambda^*| \quad and \quad d_{\mathrm{Haus}}(\check{\Lambda}, \Lambda^*) \le C \frac{k_{\mathrm{dyad}}(\theta^*)^2 \sigma^2 \log(N)}{\kappa^2}.$$

*Proof.* The proof is conducted in the events $\cap_{i \in [5]}\Omega_i \cap \mathcal{A}_1 \cap \mathcal{A}_2$, where $\Omega_1$ is defined in (S18), $\Omega_2$ is defined in (S22), $\Omega_3$ is defined in (S24), $\Omega_4$ is defined in (S26), $\Omega_5$ is defined in (S28), $\mathcal{A}_1$ is defined in (S1) and $\mathcal{A}_2$ is defined in (S2). For any $A \in \Lambda^*$, define $I_A = \{j \in [k(\tilde{\theta})] \setminus \mathcal{I} : R_j \cap A = S_j\}$.

**Step 1.** Due to (S9) and (S11), we have that $\mathcal{I} \neq \emptyset$, which implies that $I_A \neq [k(\tilde{\theta})]$ and there exists $i \notin I_A$. Let $i \notin I_A$.

**Step 1.1.** First, we claim that it is impossible that $R_i \cap A \subset S_i$ and $R_i \cap A \neq S_i$, with $\lfloor R_i \rfloor \ge \eta$. Arguing by contradiction, assume that there exists $B \in \Lambda^* \backslash \{A\}$ such that $R_i \cap B \subset S_i$ and $\bar{\theta}_A^* = \bar{\theta}_B^*$. Set

$$\mathcal{T} = \left\{B \in \Lambda^* \backslash \{A\} : R_i \cap B \subset S_i, \ \bar{\theta}_A^* = \bar{\theta}_B^*\right\},$$

and let $p \in R_i \cap A$, $q \in \bigcup_{B \in \mathcal{T}} R_i \cap B$ be such that

$$\|p - q\| = \min_{\tilde{p} \in A, \ \tilde{q} \in \bigcup_{B \in \mathcal{T}} R_i \cap B} \|\tilde{p} - \tilde{q}\|.$$

Then from Assumption S1 we have that

$$\|p - q\| \ge c \frac{k_{\mathrm{dyad}}(\theta^*)^2 \sigma^2 \log(N)}{\kappa^2}. \tag{S12}$$

Let $r^1, \ldots, r^d \in L_{d,n}$ such that for $a \in \{1, \ldots, d\}$,

$$r_b^a = \begin{cases} p_b & \text{if } b \neq a, \\ q_b & \text{if } b = a. \end{cases}$$

By construction we have that $r^1, \ldots, r^d \in R_i$. Furthermore, from (S12) there exists a $a_0 \in \{1, \ldots, d\}$ such that

$$\|p - r^{a_0}\| \geq c \frac{k_{\mathrm{dyad}}(\theta^*)^2 \sigma^2 \log(N)}{d^{1/2}\kappa^2}. \tag{S13}$$

By the definitions of $p$ and $q$, it holds that

$$\{\lambda p + (1 - \lambda)r^{a_0} \, : \, \lambda \in (0,1)\} \cap L_{d,n} \subset R_i \backslash S_i.$$

It then follows from (S13) that

$$|R_i \backslash S_i| \geq |\{\lambda p + (1 - \lambda)r^{a_0} \, : \, \lambda \in (0,1)\} \cap L_{d,n}| \geq c \frac{k_{\mathrm{dyad}}(\theta^*)^2 \sigma^2 \log(N)}{\kappa^2},$$

which contradicts the definition of $\mathcal{A}_2$.

**Step 1.2.** If $R_i \cap A = S_i$, then $|R_i| \leq \eta$.

**Step 1.3.** If $R_i \cap A \neq S_i$ and $|R_i| \geq \eta$, then by **Step 1.1**, it holds that $R_i \cap A \subset R_i \backslash S_i$. Hence, since $I_A$ induces a connected sub-graph of $G_{\mathrm{naive}}$, a fact proved below in **Step 3.**, we obtain that $R_i \cap A = \emptyset$.

**Step 2.** We claim that $I_A \neq \emptyset$. Proceeding again by contradiction, we assume that for any $j \in [k(\tilde{\theta})]$ with $R_j \cap A \neq \emptyset$, it is either the case that $|R_j| \leq \eta$ or the case that $R_j \cap A \neq S_j$. It follows from **Step 1.1.** that it is impossible to have $R_j \cap A \subset S_j$, $R_j \cap A \neq S_j$ and $|R_j| \geq \eta$. Thus, we obtain that

$$|A| \leq \sum_{j \, : \, |R_j| \leq \eta} |R_j| + \sum_{j=1}^{k(\tilde{\theta})} |R_j \backslash S_j| \leq k_{\mathrm{dyad}}(\theta^*)\eta + \frac{k_{\mathrm{dyad}}(\theta^*)\sigma^2 \log(N)}{\kappa^2}, \tag{S14}$$

where the second inequality holds due to the definitions of $\Omega_2$ and $\mathcal{A}_2$. Since $|A| \geq \Delta$, (S14) along with we constraint (S11) lead to a contradiction.

**Step 3.** We then claim that $I_A$ induces a connected sub-graph of $G_{\mathrm{naive}}$. To see this, suppose that $\{J_u\}_{u \in [l]}$ are the connected components of $I_A$ with the edges induced by $E$ and with $l > 1$. Then, if $i \in J_a$ and $j \in J_b$ for $a, b \in [l]$, $a \neq b$, then it must be the case that $\mathrm{dist}(R_i, R_j) \geq \gamma$. Hence, $\mathrm{dist}(R_i \cap A, R_j \cap A) \geq \gamma$ for all $i \in J_a$, $j \in J_b$, $a, b \in [l]$, $a \neq b$. Since $A$ is connected in $L_{d,n}$ we obtain that $|A \backslash \cup_{a=1}^l \cup_{i \in J_a} (R_i \cap A)| \geq \gamma$. However,

$$
\begin{aligned}
|A \backslash \cup_{a=1}^l \cup_{i \in J_a} R_i \cap A| &\leq \sum_{j \, : \, |R_j| \leq \eta} |R_j| + \sum_{j=1}^{k(\tilde{\theta})} |R_j \backslash S_j| \\
&\lesssim k_{\mathrm{dyad}}(\theta^*)\eta + \frac{k_{\mathrm{dyad}}(\theta^*)\sigma^2 \log(N)}{\kappa^2} < \gamma,
\end{aligned} \tag{S15}
$$

where the second inequality holds due to the definitions of $\Omega_2$ and $\mathcal{A}_1$. Thus, we have arrived at a contradiction. For any $i \in [k(\tilde{\theta})]$, let $\check{A} \in \check{\Lambda}$ be $i \in \check{A}$. We then have $R_i \subset \hat{A}$.

**Step 4.** For any $(i, j) \in [k(\tilde{\theta})] \times [k(\tilde{\theta})]$, we discuss the following two cases.

**Case 1.** If $i, j \in I_A$, then (S27) holds by the definition of $\mathcal{A}_1$, and (S28) holds by the definition of $\Omega_5$. Hence, if $\mathrm{dist}(R_i, R_j) \leq \gamma$ then $(i, j) \in E$.

**Case 2.** If $i \in I_A$ and $j \in I_B$ for $B \in \Lambda^* \backslash \{A\}$ with $\bar{\theta}_A^* \neq \bar{\theta}_B^*$, then by the definition of $\Omega_4$, we have that

$$\frac{|R_i||R_j|}{|R_i| + |R_j|} \left(\bar{Y}_{R_i} - \bar{Y}_{R_j}\right)^2 \geq \frac{\eta}{4}\kappa^2 - C_\gamma k_{\mathrm{dyad}}(\theta^*)\sigma^2 \log(N) \geq C\lambda_2,$$

provided that (S11) holds for large enough $c_1$ and $\lambda_2 = C_2 k_{\mathrm{dyad}}(\theta^*)\sigma^2 \log(N)$ for an appropriate constant. It follows that $\{i, j\} \notin E$.

**Step 5.** Combining all of the above we obtain that $|\Lambda^*| = |\tilde{\Lambda}|$. Let $\check{A} \in \check{\Lambda}$ be

$$\check{A} \in \underset{B \in \check{\Lambda}}{\arg \min} \, |B \triangle A|.$$

We have that

$$
\begin{aligned}
|\breve{A} \backslash A| &\leq \sum_{j:\,|R_j|\leq \eta} |R_j| + \sum_{j\in I_A} |R_j \backslash S_j| \leq k(\tilde{\theta})\eta + \sum_{j=1}^{k(\tilde{\theta})} |R_j \backslash S_j| \\
&\leq k_{\mathrm{dyad}}(\theta^*)\eta + C_3 \frac{k_{\mathrm{dyad}}(\theta^*)\sigma^2 \log(N)}{\kappa^2},
\end{aligned}
$$

where the last inequality holds by the definitions of $\Omega_2$ and $\mathcal{A}_1$; and

$$
\begin{aligned}
|A \backslash \breve{A}| &\leq \sum_{j:\,|R_j|\leq \eta} |R_j| + \sum_{j:\,R_j \cap A \notin \{\emptyset, S_j\}} |R_j \backslash S_j| \leq k(\tilde{\theta})\eta + \sum_{j=1}^{k(\tilde{\theta})} |R_j \backslash S_j| \\
&\leq k_{\mathrm{dyad}}(\theta^*)\eta + C_3 \frac{k_{\mathrm{dyad}}(\theta^*)\sigma^2 \log(N)}{\kappa^2}.
\end{aligned}
$$

We therefore conclude the proof. $\qquad\square$

## S5 Auxiliary results

**Noise assumption.** In the paper we make the assumption of Gaussian i.i.d. errors in (1), just like in [4]. This is a technical condition required to justify the use of Gaussian concentration inequality for Lipschitz functions. It may be relaxed by assuming errors with, e.g., log-concave density. Furthermore, it is possible to consider sub-Gaussian errors but this would involve extra logarithmic factors in the assumptions and upper bound.

Additional lemmas are collected here. Lemmas S2 and S3 follow exactly from [12], so we omit their proofs.

**Lemma S2** (Lemma 5 in [12]). *Let $I, J \subset L_{d,n}$ with $I \cap J = \emptyset$ and let $Y \in \mathbb{R}^{L_{d,n}}$. Then*

$$
\sum_{i\in I\cup J} (Y_i - \bar{Y}_{I\cup J})^2 = \sum_{i\in I}(Y_i - \bar{Y}_I)^2 + \sum_{i\in J}(Y_i - \bar{Y}_J)^2 + \frac{|I||J|}{|I|+|J|}\left(\bar{Y}_I - \bar{Y}_J\right)^2.
$$

**Lemma S3** (Lemma 6 in [12]). *Let $\mathcal{I}$ be the set of rectangles that are subsets of $L_{d,n}$. Then for $y \in \mathbb{R}^{L_{d,n}}$ defined in (1), the event*

$$
\mathcal{B} = \left\{ \max_{I,J\in\mathcal{I},\, I\cap J=\emptyset} \sqrt{\frac{|I|\,|J|}{|I|+|J|}}\, |\bar{Y}_I - \bar{\theta}_I^* - \bar{Y}_J + \bar{\theta}_J^*| \leq C_{\mathcal{B}}\sigma\sqrt{\log(N)} \right\} \tag{S16}
$$

*holds with probability at least $1 - N^{-c_{\mathcal{B}}}$, where $C_{\mathcal{B}}$ is a large enough constant and $c_{\mathcal{B}}$ depends on $C_{\mathcal{B}}$.*

**Lemma S4.** *Let $R \subset L_{d,n}$ be a rectangle and denote by $\mathcal{P}_{\mathrm{dyadic},\mathrm{d},\mathrm{n}}(R)$ the set of all dyadic partitions of $R$. Define $\beta_R \in \mathbb{R}^R$ as $\beta_R = \widetilde{\Pi}_R(y)$ where*

$$
\widetilde{\Pi}_R \in \operatorname*{arg\,min}_{\Pi\in\mathcal{P}_{\mathrm{dyadic},\mathrm{d},\mathrm{n}}(R)} \left\{ \frac{1}{2}\|y_R - O_{S(\Pi)}(y_R)\|^2 + \lambda|\Pi| \right\}. \tag{S17}
$$

*Then there exist positive constants $c_1$ and $c_2$ that depend on $d$ such that if $\lambda = C\sigma^2 \log(N)$ for a large enough constant $C > 0$ it follows that the event*

$$
\Omega_1 = \left\{ \max_{R\subset L_{d,n},\, R\,\mathrm{rectangle}} \{\|\beta_R - \theta_R^*\|^2 - 4\lambda k_{\mathrm{dyad}}(\theta_R^*)\} \leq c_1\sigma^2 \log(N) \right\} \tag{S18}
$$

*holds with probability at least $1 - N^{-c_2}$.*

*Proof.* First, proceeding as in the proof of Theorem 8.1 in [4], we obtain that

$$
\|\beta_R - \theta_R^*\|^2 \leq 2\lambda k_{\mathrm{dyad}}(\theta_R^*) + 2(y_R - \theta_R^*)^\top(\beta_R - \theta_R^*) - 2\lambda k_{\mathrm{dyad}}(\beta_R)
$$

$$
\leq 2\lambda k_{\mathrm{dyad}}(\theta_R^*) + \frac{1}{2}\|\beta_R - \theta_R^*\|^2 + 2\left\{ (y_R - \theta_R^*)^\top \frac{(\beta_R - \theta_R^*)}{\|\beta_R - \theta_R^*\|} \right\}^2 - 2\lambda k_{\mathrm{dyad}}(\beta_R). \tag{S19}
$$

Next, we denote by $\mathcal{S}_R$ the collection of linear subspaces of $\mathbb{R}^R$ such that every $S \in \mathcal{S}_R$ is a linear subspace of $\mathbb{R}^R$ such that there is a partition of $R$ and $S$ consists of piecewise constant signals over this partition of $R$. Then

$$\frac{1}{2}\|\beta_R - \theta_R^*\|^2 - 2\lambda k_{\text{dyad}}(\theta_R^*)$$

$$\leq \max_{k \in [|R|]} \sup_{S \in \mathcal{S}_R, \, \text{Dim}(S)=k} \sup_{v \in S, \, v \neq \theta_R^*} \left\{ 2 \left\{ (y_R - \theta_R^*)^\top \frac{(v - \theta_R^*)}{\|v - \theta_R^*\|} \right\}^2 - 2\lambda k_{\text{dyad}} \right\}. \tag{S20}$$

However, from Lemma 9.1 in [4], for any $c_1 > 1$, $S \in \mathcal{S}_R$ with $\dim(S) = k \in [|R|]$, we have that

$$\mathbb{P}\left( \sup_{v \in S, v \neq \theta_R^*} \left\{ 2 \left\{ (y_R - \theta_R^*)^\top \frac{(v - \theta_R^*)}{\|v - \theta_R^*\|} \right\}^2 - 2\lambda k \right\} \geq c_1 \sigma^2 \log(N) \right)$$

$$\leq \mathbb{P}\left( \sup_{v \in S, v \neq \theta_R^*} 2 \left\{ (y_R - \theta_R^*)^\top \frac{(v - \theta_R^*)}{\|v - \theta_R^*\|} \right\}^2 \geq c_1 \sigma^2 \log(N) + 2\lambda k \right)$$

$$\leq 2 \exp\left( -\frac{c_1/2 \log(N) + (2\lambda/\sigma^2 - 2)k - 4}{8} \right).$$

Since $|\{S \in \mathcal{S}_R \, : \, \text{Dim}(S) = k\}| \leq |R|^{2k}$, it follows by a union bound argument that for some $c_2 > 0$,

$$\mathbb{P}\left( \sup_{S \in \mathcal{S}_R, \, \text{Dim}(S)=k} \sup_{v \in S, v \neq \theta_R^*} \left\{ 2 \left\{ (y_R - \theta_R^*)^\top \frac{(v - \theta_R^*)}{\|v - \theta_R^*\|} \right\}^2 - 2\lambda k \right\} \geq c_1 \sigma^2 \log N \right)$$

$$\leq \exp\left( -c_2 \log(N) \right), \tag{S21}$$

provided that $\lambda = C\sigma^2 \log(N)$ with a sufficiently large $C > 0$. The claim follows from a union bound argument by combining (S20), (S21), the fact that there are most $N^2$ subrectangles of $L_{d,n}$, and choosing $c_1$ large enough. $\square$

**Lemma S5.** *Let $\tilde{\theta}$ be the DCART estimator. If $\lambda = C\sigma^2 \log(N)$ for a large enough $C$, then there exist positive constants $c_3$ and $c_4$ such that the event*

$$\Omega_2 = \left\{ k(\tilde{\theta}) \leq 2k_{\text{dyad}}(\theta^*) + c_3 \right\} \tag{S22}$$

*holds with probability at least $1 - N^{-c_4}$.*

*Proof.* First notice that by the basic inequality (S19), it holds that

$$\lambda k(\tilde{\theta}) \leq 2\lambda k_{\text{dyad}}(\theta^*) + 2 \left\{ (y - \theta^*)^\top \frac{(\tilde{\theta} - \theta^*)}{\|\tilde{\theta} - \theta^*\|} \right\}^2 - \lambda k(\tilde{\theta}). \tag{S23}$$

Therefore, from Lemma S4, choosing $\lambda = C\sigma^2 \log(N)$ with large enough $C$ implies that with probability at least $1 - N^{-c_2}$ the event

$$\Omega = \left\{ 2 \left\{ (y - \theta^*)^\top \frac{(\tilde{\theta} - \theta^*)}{\|\tilde{\theta} - \theta^*\|} \right\}^2 - \lambda k(\tilde{\theta}) \geq c_1 \sigma^2 \log(N) \right\}.$$

holds. Considering (S23) on the event $\Omega$, we have that

$$k(\tilde{\theta}) \leq 2k_{\text{dyad}}(\theta^*) + \frac{2c_1}{C}$$

and the claim follows. $\square$

**Lemma S6.** *The event*

$$\Omega_3 = \left\{ \max_{R \subset L_{d,n}, \, R \text{ rectangle}} |R| \, |\bar{\theta}_R^* - \bar{y}_R|^2 \leq c_1 \sigma^2 \log N \right\} \tag{S24}$$

*holds with probability at least $1 - N^{-c_2}$ for some postive constants $c_1$ and $c_2$.*

*Proof.* This follows immediately from the fact that there are at most $N^2$ rectangles, the Gaussian tail inequality and a union bound argument. □

**Lemma S7.** *With the notation of Theorem 1, we define the set $\mathcal{Q}_4 \subset [k(\tilde{\theta})] \times [k(\tilde{\theta})]$ as*

$$\mathcal{Q}_4 = \left\{ (i,j) : \bar{\theta}^*_{S_i} \neq \bar{\theta}^*_{S_j}, \ |R_i| \leq 2|S_i|, \ |R_j| \leq 2|S_j| \right\}. \tag{S25}$$

*Define the event*

$$\Omega_4 = \left\{ \frac{|R_i| |R_j|}{|R_i| + |R_j|} \left( \bar{Y}_{R_i} - \bar{Y}_{R_j} \right)^2 \geq \frac{\min\{|R_i|, |R_j|\}}{4} \kappa^2 - C k_{\mathrm{dyad}}(\theta^*) \sigma^2 \log(N), \ \forall (i,j) \in \mathcal{Q}_4 \right\}, \tag{S26}$$

*where $C > 0$ is an absolute constant. Then there exists an absolute constant $c > 0$ such that the event $\Omega_4$ holds with probability at least $1 - N^{-c}$.*

*Proof.* The proof is conducted assuming the high-probability event $\mathcal{B}$ defined in (S16). Now, any for $(i,j) \in \mathcal{Q}_4$, we have that

$$\frac{|R_i| |R_j|}{|R_i| + |R_j|} \left( \bar{Y}_{R_i} - \bar{Y}_{R_j} \right)^2 = \frac{|R_i| |R_j|}{|R_i| + |R_j|} \left\{ -\bar{\theta}^*_{R_j} + \bar{\theta}^*_{R_i} + ( \bar{Y}_{R_i} - \bar{\theta}^*_{R_i} + \bar{\theta}^*_{R_j} - \bar{Y}_{R_j} ) \right\}^2$$

$$\geq \frac{|R_i| |R_j|}{2(|R_i| + |R_j|)} \left( -\bar{\theta}^*_{R_j} + \bar{\theta}^*_{R_i} \right)^2 - \frac{|R_i| |R_j|}{|R_i| + |R_j|} \left( \bar{Y}_{R_i} - \bar{\theta}^*_{R_i} + \bar{\theta}^*_{R_j} - \bar{Y}_{R_j} \right)^2$$

$$\geq \frac{|R_i| |R_j|}{2(|R_i| + |R_j|)} \left( -\bar{\theta}^*_{R_j} + \bar{\theta}^*_{R_i} \right)^2 - C^2_{\mathcal{B}} \sigma^2 \log(N)$$

$$= \frac{|R_i| |R_j|}{2(|R_i| + |R_j|)} \left\{ \bar{\theta}^*_{S_i} - \bar{\theta}^*_{S_j} + (\bar{\theta}^*_{S_j} - \bar{\theta}^*_{R_j} + \bar{\theta}^*_{R_i} - \bar{\theta}^*_{S_i}) \right\}^2 - C^2_{\mathcal{B}} \sigma^2 \log(N)$$

$$\geq \frac{|R_i| |R_j|}{4(|R_i| + |R_j|)} \left( \bar{\theta}^*_{S_i} - \bar{\theta}^*_{S_j} \right)^2 - \frac{|R_i| |R_j|}{2(|R_i| + |R_j|)} \left( \bar{\theta}^*_{S_j} - \bar{\theta}^*_{R_j} + \bar{\theta}^*_{R_i} - \bar{\theta}^*_{S_i} \right)^2$$
$$- C^2_{\mathcal{B}} \sigma^2 \log(N)$$

$$\geq \frac{|R_i| |R_j|}{4(|R_i| + |R_j|)} \left( \bar{\theta}^*_{S_i} - \bar{\theta}^*_{S_j} \right)^2 - \frac{|R_i| |R_j|}{|R_i| + |R_j|} \left( \bar{\theta}^*_{S_j} - \bar{\theta}^*_{R_j} \right)^2 - \frac{|R_i| |R_j|}{|R_i| + |R_j|} \left( \bar{\theta}^*_{S_i} - \bar{\theta}^*_{R_i} \right)^2$$
$$- C^2_{\mathcal{B}} \sigma^2 \log(N)$$

$$\geq \frac{\min\{|R_i|, |R_j|\}}{4} \kappa^2 - |R_j| \left( \bar{\theta}^*_{S_j} - \bar{\theta}^*_{R_j} \right)^2 - |R_i| \left( \bar{\theta}^*_{S_i} - \bar{\theta}^*_{R_i} \right)^2 - C^2_{\mathcal{B}} \sigma^2 \log(N)$$

$$\geq \frac{\min\{|R_i|, |R_j|\}}{4} \kappa^2 - |R_j| \left\{ \frac{1}{|S_j|} \sum_{l \in S_j} (\theta^*_l - \bar{\theta}^*_{R_j})^2 \right\} - |R_i| \left\{ \frac{1}{|S_i|} \sum_{l \in S_i} (\theta^*_l - \bar{\theta}^*_{R_i})^2 \right\}$$
$$- C^2_{\mathcal{B}} \sigma^2 \log(N),$$

where the first and third inequalities follow from the inequality $(a+b)^2 \geq a^2/2 - b^2$, the second by the definition of $\mathcal{B}$ in (S16), the fourth by the inequality $(a+b)^2 \leq 2a^2 + 2b^2$ and the sixth by Jensen's inequality. Then in the event $\Omega_1$ defined in (S18), it from Lemma S4 that,

$$\frac{|R_i| |R_j|}{|R_i| + |R_j|} \left( \bar{Y}_{R_i} - \bar{Y}_{R_j} \right)^2$$

$$\geq \frac{\min\{|R_i|, |R_j|\}}{4} \kappa^2 - 2 \sum_{l \in S_j} (\theta^*_l - \bar{\theta}^*_{R_j})^2 - 2 \sum_{l \in S_i} (\theta^*_l - \bar{\theta}^*_{R_i})^2 - C^2_{\mathcal{B}} \sigma^2 \log(N)$$

$$\geq \frac{\min\{|R_i|, |R_j|\}}{4} \kappa^2 - 4 \sum_{l \in S_j} (\theta^*_l - \bar{y}_{R_j})^2 - 4|S_j| (\bar{y}_{R_j} - \bar{\theta}^*_{R_j})^2$$

$$- 4 \sum_{l \in S_i} (\theta^*_l - \bar{y}_{R_i})^2 - 4|S_i| (\bar{y}_{R_i} - \bar{\theta}^*_{R_i})^2 - C^2_{\mathcal{B}} \sigma^2 \log(N)$$

$$\geq \frac{\min\{|R_i|, |R_j|\}}{4} \kappa^2 - C \sigma^2 k_{\mathrm{dyad}}(\theta^*) \log(N),$$

for some constant $C > 0$, where the second inequality follows from the inequality $(a + b)^2 \leq 2a^2 + 2b^2$, and the last one by Lemmas S4 and S6. The claim then follows. $\qquad\square$

**Lemma S8.** *With the notation of Theorem 1, we define the set $\mathcal{Q}_5 \subset [k(\tilde{\theta})] \times [k(\tilde{\theta})]$ as*

$$\mathcal{Q}_5 = \left\{ (i,j) : \bar{\theta}^*_{S_i} = \bar{\theta}^*_{S_j}, \, |R_i| \leq 2|S_i|, \, |R_j| \leq 2|S_j| \right\}. \tag{S27}$$

*Define the event*

$$\Omega_5 = \left\{ \frac{|R_i| \, |R_j|}{|R_i| + |R_j|} \left( \bar{Y}_{R_i} - \bar{Y}_{R_j} \right)^2 \leq C k_{\mathrm{dyad}}(\theta^*) \sigma^2 \log(N), \, \forall (i,j) \in \mathcal{Q}_5 \right\}, \tag{S28}$$

*where $C > 0$ is an absolute constant. Then there exists an absolute constant $c > 0$ such that $\Omega_5$ holds with probability at least $1 - N^{-c}$.*

*Proof.* We assume through that that the high-probability event $\mathcal{B}$ defined in (S16) holds. Let $(i,j) \in \mathcal{Q}_5$. Then,

$$
\begin{aligned}
& \frac{|R_i| \, |R_j|}{|R_i| + |R_j|} \left( \bar{Y}_{R_i} - \bar{Y}_{R_j} \right)^2 \\
\leq \; & \frac{2|R_i| \, |R_j|}{|R_i| + |R_j|} \left( \bar{Y}_{R_i} - \bar{\theta}^*_{R_i} - \bar{Y}_{R_j} + \bar{\theta}^*_{R_j} \right)^2 + \frac{2|R_i| \, |R_j|}{|R_i| + |R_j|} \left( \bar{\theta}^*_{R_j} - \bar{\theta}^*_{R_i} \right)^2 \\
\leq \; & 2 C_{\mathcal{B}} \sigma^2 \log(N) + \frac{4|R_i| \, |R_j|}{|R_i| + |R_j|} \left( \bar{\theta}^*_{R_j} - \bar{\theta}^*_{S_j} \right)^2 + \frac{4|R_i| \, |R_j|}{|R_i| + |R_j|} \left( \bar{\theta}^*_{R_i} - \bar{\theta}^*_{S_i} \right)^2 \\
\leq \; & 2 C_{\mathcal{B}} \sigma^2 \log(N) + \frac{4|R_j|}{|S_j|} \sum_{l \in S_j} (\theta^*_l - \bar{\theta}^*_{R_j})^2 + \frac{4|R_i|}{|S_i|} \sum_{l \in S_i} (\theta^*_l - \bar{\theta}^*_{R_i})^2 \\
\leq \; & 2 C_{\mathcal{B}} \sigma^2 \log(N) + 8 \sum_{l \in S_j} (\theta^*_l - \bar{\theta}^*_{R_j})^2 + 8 \sum_{l \in S_i} (\theta^*_l - \bar{\theta}^*_{R_i})^2 \\
\leq \; & 2 C_{\mathcal{B}} \sigma^2 \log(N) + 16 \sum_{l \in S_j} (\theta^*_l - \bar{y}_{R_j})^2 + 16 \sum_{l \in S_i} (\theta^*_l - \bar{y}_{R_i})^2 \\
& + 16 |S_j| (\bar{\theta}^*_{R_j} - \bar{y}_{R_j})^2 + 16 |S_i| (\bar{\theta}^*_{R_i} - \bar{y}_{R_i})^2.
\end{aligned}
$$

The first and second inequalities use the trivial fact that $(a + b)^2 \leq 2a^2 + 2b^2$, the second inequality uses the event $\mathcal{B}$ and the third follows from Lemma S2. Combining the above inequality with Lemmas S3, S4 and S6 completes the proof. $\qquad\square$

## S6 Experiments section details

### S6.1 Scenarios

We detail all the signal patterns considered in the simulations in Section 3. All these scenarios are depicted in Figure S1.

**Scenario 1**. For all $(a, b) \in L_{2,n}$, let

$$
\theta^*_{(a,b)} = \begin{cases} 1 & \text{if } \frac{n}{4} < a < \frac{3n}{4} \text{ and } \frac{n}{4} < b < \frac{3n}{4}, \\ 0 & \text{otherwise.} \end{cases}
$$

**Scenario 2**. For all $(a, b) \in L_{2,n}$, let

$$
\theta^*_{(a,b)} = \begin{cases} 1 & \text{if } (a - \frac{n}{4})^2 + (b - \frac{n}{4})^2 < \left(\frac{n}{5}\right)^2, \\ 1 & \text{if } (a - \frac{3n}{4})^2 + (b - \frac{3n}{4})^2 < \left(\frac{n}{5}\right)^2, \\ 0 & \text{otherwise.} \end{cases}
$$

**Scenario 3**. For all $(a, b) \in L_{2,n}$, let

$$
\theta^*_{(a,b)} = \begin{cases} 1 & \text{if } a \in (\frac{n}{4}, \frac{3n}{4}) \text{ and } b \in (\frac{n}{4}, \frac{3n}{8}), \\ 1 & \text{if } a \in (\frac{5n}{8}, \frac{3n}{4}) \text{ and } b \in [\frac{3n}{8}, \frac{3n}{4}), \\ -1 & \text{if } a > \frac{3n}{4} \text{ and } b > \frac{3n}{4}, \\ 0 & \text{otherwise.} \end{cases}
$$

**Scenario 4.** For all $(a, b) \in L_{2,n}$, let

$$
\theta^*_{(a,b)} = \begin{cases}
1 & \text{if } a < \frac{n}{5} \text{ and } b < \frac{n}{5}, \\
2 & \text{if } a < \frac{n}{5} \text{ and } b > \frac{4n}{5}, \\
3 & \text{if } a > \frac{4n}{5} \text{ and } b < \frac{4n}{5}, \\
4 & \text{if } a > \frac{4n}{5} \text{ and } b > \frac{4n}{5}, \\
5 & \text{if } a \in (\frac{3n}{8}, \frac{5n}{8}) \text{ and } b \in (\frac{3n}{8}, \frac{5n}{8}), \\
0 & \text{otherwise.}
\end{cases}
$$

## S6.2 Tuning parameters for naive two step-estimator

We first construct a sequence of DCART estimators $\tilde{\theta}(\lambda)$, $\lambda \in \mathcal{S}_\lambda = \{5 + (30 - 5)l/14, \, l = 0, \ldots, 14\}$. Indexing the nodes in $L_{d,n}$ as $\{i_1, \ldots, i_{n^2}\}$, we calculate

$$
\hat{\sigma}^2 = (2n^2)^{-1} \sum_{j \in [n^2 - 1]} (y_{i_j} - y_{i_{j+1}})^2.
$$

Based on this variance estimator, we choose

$$
\lambda_1 = \underset{\lambda \in \mathcal{S}_\lambda}{\arg\min} \left[ \sum_{i \in L_{d,n}} \{y_i - \tilde{\theta}_i(\lambda)\}^2 + \hat{\sigma}^2 k(\tilde{\theta}(\lambda)) \log(N) \right] \quad \text{and} \quad \tilde{\theta} = \tilde{\theta}(\lambda_1).
$$

Once $\tilde{\theta}$ is computed, in the second step, we construct the final estimator denoted here as $\widehat{\Lambda}$ by setting $\lambda_2 = \lambda_1$, $\gamma = 2^3$ and $\eta = 2^3$ (see Section S4). The choice $\lambda_2 = \lambda_1$ is consistent with the theory, since in all the scenarios considered here $k_{\text{dyad}}(\theta^*)$ is small.

## S6.3 Implementation details of total variation based estimator

We now discuss the implementation details for the total variation based estimator used in our experiments. Starting from the $L_{d,n}$ lattice, we let $D$ be an incidence matrix corresponding to $L_{d,n}$, see for instance [9]. We then compute, using the algorithm from [8], the estimators

$$
\beta_\lambda = \underset{\beta \in \mathbb{R}^{L_{d,n}}}{\arg\min} \left\{ \frac{1}{2}\|\beta - y\|^2 + \lambda \|D\beta\|_1 \right\}
$$

for $\lambda \in \{10^{3l/19} : l = 0, 1, \ldots, 19\}$. Then letting $\hat{\sigma}^2$ as in Section 3, we let

$$
\lambda^* = \underset{\lambda \in \{10^{3l/19} : l = 0,1,\ldots,19\}}{\arg\min} \left\{ \|\beta_\lambda - y\|^2 + \hat{\sigma}^2 c(\beta_\lambda) \log(N) \right\}
$$

where $c(\beta_\lambda)$ is the number of connected components in $L_{d,n}$ induced by $\beta_\lambda$. In other words, $c(\beta_\lambda)$ is the estimated degrees of freedom in the model associated with $\beta_\lambda$ in the language of [10]. Then we set $\hat{\beta}$ equal to $\beta_{\lambda^*}$ after rounding each entry of $\beta_{\lambda^*}$ to three decimal digits.

Next, let $\{R_l\}_{l \in [q]}$ be the partition of $L_{d,n}$ induced by $\hat{\beta}$, $\eta = \gamma = 8$ and $a = 0.15$. For each $(i, j) \in [q] \times [q]$, let $z_{(i,j)} = 1$ if

$$
\text{dist}(R_i, R_j) \leq \gamma, \quad \min\{|R_i|, |R_j|\} \geq \eta
$$

and $|\bar{Y}_{R_i} - \bar{Y}_{R_j}| < a$; otherwise, let $z_{(i,j)} = 0$. With this notation, let $E = \{e \in [q] \times [q] : z_e = 1\}$ and let $\{\mathcal{C}_l\}_{l \in [\hat{L}]}$ be the collection of all the connected components of the undirected graph $([q]\backslash\mathcal{I}, E)$, where $\mathcal{I} = \{i \in [q] : |R_i| \leq \eta\}$. Our final estimator becomes

$$
\Lambda' = \left\{ \cup_{j \in \mathcal{C}_1} R_j, \ldots, \cup_{j \in \mathcal{C}_{\hat{L}}} R_j \right\}. \tag{S29}
$$

Notice that in (S29) we do not include the sets $R_j$ with a small number of elements as we found that by using them the performance of the estimator becomes worst.

Table S1: Performance evaluations over 50 repetitions under Scenario 5. The performance metrics $\text{dist}_1$ and $\text{dist}_2$ are defined in the text. The numbers in parenthesis denote standard errors.

| Setting | | dist$_1$ | | dist$_2$ | |
|---|---|---|---|---|---|
| | $\sigma$ | $\widehat{\Lambda}$ | TV-based | $\widehat{\Lambda}$ | TV-based |
| 5 | 0.5 | 211.68(745.74) | **130.72(44.49)** | **0.0(0.27)** | 0.12(0.33) |
| 5 | 1.0 | 766.84(1281.38) | **398.92(362.39)** | **0.0(0.57)** | 1.32(0.9) |
| 5 | 1.5 | 1406.96(1589.79) | **921.08(214.55)** | **0.0(0.65)** | 2.68(1.31) |

## S6.4 Additional scenario

In this subsection we consider an additional scenario, namely Scenario 5. For all $(a,b) \in L_{2,n}$, we let

$$\theta^*_{(a,b)} = \begin{cases} 2 & \text{if } a < \frac{n}{5} \text{ and } b > \frac{2n}{5}, \\ 3 & \text{if } a > \frac{4n}{5} \text{ and } b < \frac{3n}{5}, \\ 4 & \text{if } |a - \frac{n}{2}| < \frac{n}{4.5} \text{ and } b < \frac{n}{4.5}, \\ 0 & \text{otherwise}. \end{cases}$$

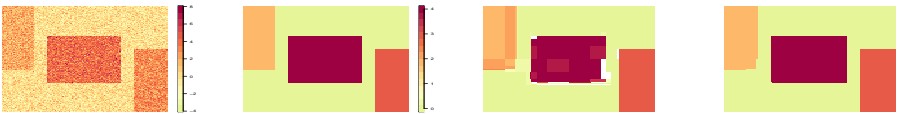

Figure S1: Visualization of Scenario 5. From left to right: An instance of $y$, the signal $\theta^*$, DCART, and DCART after merging. In this example the data are generated with $\sigma = 1$.

Performance evaluations for Scenario 5 are given in Table S1. There, we can see that our proposed method provides the best estimation of the number of piecewise constant regions.

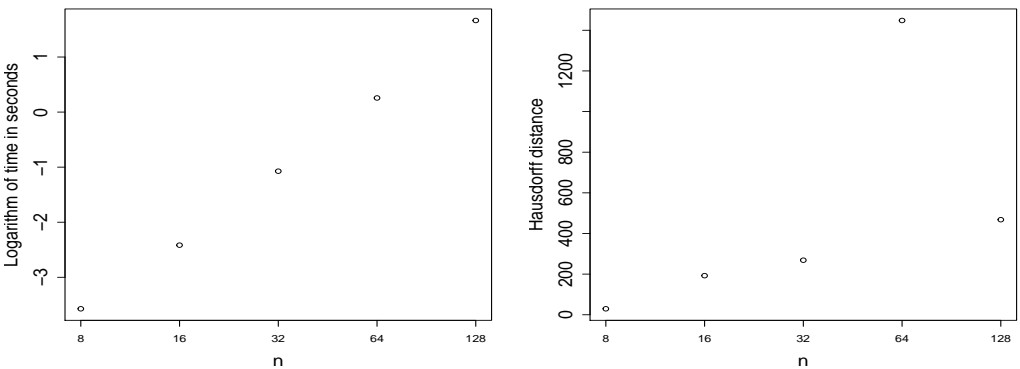

Figure S2: Time and Hausdorff distance evaluations, averaging over 50 Monte Carlo simulations, of $\widehat{\Lambda}$ for different values of $n$ for Scenario 5. Here $\sigma = 1$.

Finally, with the same implementation details as in Section 3 of the paper, we compute the running time of $\widehat{\Lambda}$ for Scenario 5. The results are shown in Figure S2 where we can clearly see a linear trend.

## S7 Non axis-aligned data

We now briefly discuss how our method could be extended to non axis-aligned data. Suppose that we are given measurements $\{(x_i, y_i)\}_{i=1}^N$ which are independent copies of a pair of random variables

$(X, Y) \in [0, 1]^d \times \mathbb{R}$. Suppose that $n \asymp (N / \log N)^{1/d}$ with $n \in \mathbb{N}$. Define

$$I_{i_1, \ldots i_d} := \left[ \frac{i_1 - 1}{n}, \frac{i_1}{n} \right] \times \ldots \times \left[ \frac{i_d - 1}{n}, \frac{i_d}{n} \right]$$

for $i_1, \ldots, i_d \in \{1, \ldots, n\}$. Then define $\tilde{y} \in \mathbb{R}^{L_{d,n}}$ as

$$\tilde{y}_{i_1, \ldots i_d} := \frac{1}{|\{j \,:\, y_j \in I_{i_1, \ldots i_d}\}|} \sum_{j \,:\, y_j \in I_{i_1, \ldots i_d}} y_j,$$

if $|\{j \,:\, y_j \in I_{i_1, \ldots i_d}\} \neq \emptyset$ and otherwise we set $\tilde{y}_{i_1, \ldots i_d} = \tilde{y}_{i'_1, \ldots i'_d}$ where $I_{i'_1, \ldots i'_d}$ is the closest rectangle to $I_{i_1, \ldots i_d}$ satisfying $|\{j \,:\, y_j \in I_{i'_1, \ldots i'_d}\} \neq \emptyset$. Both the choice of $n$ and the constrution of $\tilde{y}$ are inspired by ideas from [6].

After having constructed $\tilde{y} \in \mathbb{R}^{L_{d,n}}$ we can then run DCART and our modified version.