# OpenReview forum: "Lattice partition recovery with dyadic CART"
_NeurIPS.cc/2021/Conference — NeurIPS 2021 Poster_

### Official Review · Reviewer_ZZUp · 2021-07-11

**Rating:** 6
**Confidence:** 3

**Summary:**

This paper considers a piecewise constant signal on a $d$-dimensional lattice. The observed signal is corrupted by additive Gaussian noise. The task is to recover the partition of the lattice such that the signal within a partition is constant. The paper proposes an algorithm that builds upon the scalable DCART algorithm.

The contributions of the paper are threefold:

(i)	The authors first analyze the performance of the vanilla DCART algorithm to recover the partitions. They prove the one-sided consistency of the vanilla DCART algorithm. One-sided consistency guarantees that the recovered partitions have large subsets which have constant signal, but a partition of the ground truth signal could potentially be over-partitioned by the DCART algorithm. They demonstrate by an example that it is indeed the case.

(ii)	In order to address the issue of over-partitioning, the authors propose a two-step algorithm where the first step is the vanilla DCART algorithm while restricting to partitions with size above a threshold. In the second step, the partitions which are close and whose signals have a small difference are merged. They prove that this estimator is consistent.

(iii)	They prove that the proposed 2 step estimator is minimax optimal if the number of partitions is constant.
Finally, they demonstrate the efficacy of their algorithm by simulations.


**Limitations And Societal Impact:**

There does not seem to be any potential negative societal impact of this work.

**Main Review:**

Originality: The authors tackle the problem of partition recovery for a general dimensional lattice which (most likely) has not received attention before. An important question that authors do not address in the paper is why is it an interesting problem to recover the partition, what are the settings where having an estimate of the signal itself is not good enough? Furthermore, it is not clear how distinct is the task of partition recovery from that of signal recovery. When we have access to the exact signal, recovering the partition from the signal is easy. Similarly, one could potentially perform a simple post-processing step after signal reconstruction where one merges the neighboring lattice points with close estimated signal values to form a partition. A clearer discussion on these lines would help in motivating the problem better.

While the DCART algorithm itself is not new, its application to the task at hand and the theoretical analysis is novel. The paper clearly delineates itself from previous work and the related work is well cited.

Quality: The paper is technically sound and the proofs are clear and interesting. The authors do bring up the issue of the estimator being minimax optimal only under the restrictive setting of $k_{dyad}(\theta^*)$ being constant. I believe that this is a significant bottleneck, especially as $d$ scales. For instance, if the partitions are hypercubes, the number would grow exponentially in $d$. Thus, the suboptimality could potentially be very large.

Clarity: The paper is fairly well written. It is well organized and includes all the details.

Significance: The results are important because the paper uses the scalable DCART algorithm for a new task. It would really help the paper if more thought goes into motivating the problem.

Minor points-

(i)	I could not really understand what the second sentence of Definition 1 was trying to say. It switches from $\theta \in \mathbb{R}^{L_{d,n}}$ in the first sentence to $\theta \in {L_{d,n}}$ in the second sentence. Probably $k(\theta)$ is the smallest cardinality of $\{R_l\}$ defined in the first sentence?

(ii)	Line 327 onwards – the signal patterns are in the second column the third and fourth column depict the vanilla DCART and the 2 step estimator respectively.

Overall I really like the technical contributions of the paper but the paper is held back by the lack of motivation of the problem and the suboptimality of the estimator.


**Time Spent Reviewing:**

15

---

> ### Author Response · Authors · 2021-08-10
> **Responses to the comment on the importance of partition recovery and others.**
>
> Thank you very much for your appreciation and constructive comments.  In the following, we provide detailed responses to your comments.
>
> *  __Originality.__  Recovering the partition and therefore localizing different signals have a wide array of applications, especially in surveillance and environment monitoring.  Our work is motivated by all the applications/problems considered in the large literature on biclustering, where the underlying signal is assumed to be piecewise constant.  Estimating the boundary of the partition is the most refined and difficult task in these settings.  Thus, any of the many scenarios in which biclustering is relevant can be used to motive our task. An analogous observation holds also for the more general problem of identifying an anomalous cluster (sub-graph) in a network, a problem that has been tackled (for testing purposes only) by Arias-Castro, Candes \& Durand, A. (2011); as we mention in our paper, in that reference the authors provide numerous examples of applications.  We will add more detailed discussions on this front, should the paper be accepted.  In terms of the theoretical tasks, the relationship between the partition and signal recoveries can be thought of the relationship between the estimation consistency and support consistency in a high-dimensional linear regression problems.  They can be done by almost identical algorithms but the theoretical results rely on different sets of conditions.  What we have done in this paper is exactly what you described.  We start with the DCART, which is exploited extensively in signal recovering and the theoretical guarantee in terms of signal recovering is studied in the existing literature.  In order to achieve the optimality in partition recovery, we adopt a further merging step based on the output of the DCART.  In this paper, we focus on understanding the ability of recovering the partition of DCART and how to achieve the optimality.
>
> * __Quality.__  The method works for any fixed but arbitrary dimension $d$, but it suffers both computationally and theoretically when $d$ is large.  More tailored algorithms are necessary when $d$ is moderately large.
>
> *  __Significance.__  Thanks for your comment and suggestion.  We will add more detailed discussions should the paper be accepted.

---

> > ### Comment · Reviewer_ZZUp · 2021-09-01
> > **Response to the author's comments.**
> >
> > The response from the authors is very helpful. I believe that adding discussion on similar lines would help in not only motivating the problem well but also highlighting why the problem cannot be solved by a simple postprocessing step. However, I still could not find a clear response on the lines of - for this application, signal recovery does not work, we need to recover the partition. Furthermore, the paper is held back by the sample complexity suboptimality in $d$. Overall, I like the paper and will stick to my current evaluation of a weak accept.

---

> > > ### Author Response · Authors · 2021-09-01
> > > **Response to the updates.**
> > >
> > > Thanks for your responses.
> > >
> > > 1.  Regarding the applications, [1] contains a wealth of examples and applications and we cite the following paragraph in [1] as a motivating example.
> > >
> > > "Disease outbreak detection. The presence of a biological or chemical material in a given geographical region may also be detected indirectly through its impact on human health. In this context, early detection of the disease outbreak is crucial in order to minimize the severity of the epidemic. For that purpose, some specific information networks are used, with surveillance systems now incorporating data from hospital emergency visits, ambulance dispatch calls and pharmacy sales of over-the-counter drugs."
> > >
> > > Note that, the statistical inference goal in [1] is to detect the existence of an outbreak, while we focus on localising where exactly the outbreak happens.  As for the goal of signal recovery, the statistical guarantees are to upper bound the error of the whole signal estimator, which does not directly imply an optimal localisation of where the outbreak is.  For example, in the 1-d chain graph with piecewise constant mean, it is known that fused lasso is able to optimally estimate the whole signal but sub-optimally locate where the change points are.
> > >
> > > 2.  You are indeed right that the dependence on the dimensionality $d$ is not satisfactory, but it is yet known whether this is optimal or suboptimal.
> > >
> > > [1] Arias-Castro, E., Candes, E. J., & Durand, A. (2011). Detection of an anomalous cluster in a network. The Annals of Statistics, 278-304.

---

### Official Review · Reviewer_2a7D · 2021-07-16

**Rating:** 6
**Confidence:** 2

**Summary:**

The authors propose to study the use of CART to recover the partition underlying piecewise constant noisy data. They perform empirical validation and provide extensive theoretical analysis of the algorithm showing its benefits and pitfalls.

**Limitations And Societal Impact:**

yes

**Main Review:**

The main critic I have is the lack of motivation in the task of recovering the underlying partition of the data. Can this be employed to then produce better denoising for example by finding new criterion that would make the tree converge faster?

Additionally a discussion on dealing with non axis-aligned partition would have been appreciated. For example to see how good/bad can this estimator become in the worst case and what could be some simple strategies to mitigate this (like oblique trees?). This is not missing from the current manuscript but would have elevated the impact of the paper.

Also the experiments are missing cases in which the piecewise constant regions have different support sizes between regions. This seems like an important case to study especially to know which conditions the data need to follow in order to have best performances. It seems that as the number of regions grows (and their size reduces) for the true data, as the CART produced partition would potentially diverge whenever the noise std is not negligible. Also, it seems that some naive baselines (e.g. low pass denoising followed by thresholding) allowing to recover the data partition would have helped to measure the actual benefit of the proposed algorithm.

Beyond that point I believe that the manuscript is well written and overall thorough and clear.

"partition recover problem" -> "partition recovery problem"



------------- UPDATE AFTER DISCUSSION -------------

I thank the authors for their answers, I have updated my score as I believe that with those discussions, the paper would be above the acceptance level, **especially the link between partition geometry and performances**. One last thing that would be very interesting (even though probably only to be discussed in a future work/discussion section alongside with non-axis aligned data) is on how one could combine the proposed approach with decision trees that can be learned in a differentiable way (e.g. https://ieeexplore.ieee.org/document/817409, https://arxiv.org/pdf/1702.07360.pdf).

**Time Spent Reviewing:**

5

---

> ### Author Response · Authors · 2021-08-10
> **Responses to the comments on motivation and others.**
>
> Thank you very much for your appreciation and constructive comments.  In the following, we provide detailed responses to your comments.
>
> * **On the motivation.**  Firstly, there is a wide array of applications focusing on detecting the regions rather than estimating the background signals, especially in surveillance and environment monitoring.  Our work is motivated by all the applications/problems considered in the large literature on biclustering, where the underlying signal is assumed to be piecewise constant. Estimating the boundary of the partition is the most refined and difficult task in these settings. Thus, any of the many scenarios in which biclustering is relevant can be used to motive our task. An analogous observation holds also for the more general problem of identifying an anomalous cluster (sub-graph) in a network, a problem that has been tackled (for testing purposes only) by Arias-Castro, Candes & Durand, A. (2011); as we mention in our paper, in that reference the authors provide numerous examples of applications.   We will add more detailed applications and motivations should the paper be accepted.  Secondly, as the experiments show the performance of the modified estimator is better for denoising than the original DCART, see the third and fourth panels of Figure 3.  Lastly, DCART is amenable due to its interpretability, but what we are providing is an even more interpretable estimator that has fewer spurious rectangles.
>
> * **On non axis-aligned data.**  To deal with non axis-aligned data, one can construct a lattice in the domain of the features, average the observations within each cell of the lattice and ignore the cells without observations.  We will add more comments along this line should the paper be accepted.
>
> * **On the numerical experiments.**  Thank you very much for your suggestion.  We will include more experiments on this front in the revision.  In fact, as the support sizes grow larger, the performance of both the original DCART and our modified version would deteriorate.

---

> ### Author Response · Authors · 2021-08-30
> **Responses to the updates**
>
> We thank the reviewer for the positive comments and for pointing out the very intriguing connection to differentiable decision trees and the relevant literature.  We will include them in the discussion as a direction for future work.

---

### Official Review · Reviewer_qw8i · 2021-07-18

**Rating:** 7
**Confidence:** 4

**Summary:**

This paper studies a problem called lattice partition recovery, where piecewise constant signals over a d-dimensional lattice are corrupted by noise and require to be recovered. A method called dyadic classification and regression tree (DCART) is studied. It is shown that DCART achieves one-sided consistency, but suffers from over-partitioning. To achieve partition recovery, this paper proposes an additional merging stage, where dyadic rectangles that are close are merged. This is shown to achieve partition recovery under mild conditions. A minimax lower bound is also given, and numerical experiments are shown to support the methodology.

**Main Review:**

This paper studies a very interesting and fundamental statistical estimation problem. It appears that this paper provides a natural improvement of DCART so that partition recovery is achieved. It is further supported by solid theory and experiments. I think that this paper could make a potential impact in the statistical and machine learning community.

However, there is certain ambiguity in the description of the method. This may be caused by the page limit, but I do hope the authors would clarify some points from a possibly quick revision.

- What is exactly the space of partitions in the optimization problem (4) and (8)? Does the space contain all the possible choices of rectangle $R_u$ in each split? If so, then the space can be quite large, is that correct? What is the size of the space?

- Following the previous question, how do we find the optimizer of (4) and (8) algorithmically? Do we do greedy splits, or do we search the space exhaustively? The reference [14] says the algorithm has complexity $O(n^2)$ in 2-D grid. So here the algorithm has complexity $O(n^d)$?

- On line 178--179, it is unclear what it means by "instead of DCART, it is possible to deploy the ORT estimator [12] or the NP-hard estimator (2) in our algorithms".

- It seems to me, intuitively, that the penalty $\lambda_1 | \Pi |$ in (8) already discourages over-partitioning. Why do we need the additional merging step to achieve partition recovery?

- In the merging stage, we merge, say, rectangles $R_i$ and $R_j$. From (9), they are close but they are not required to be neighbors. Is that a bit strange?

- In Theorem 2, I think there are typos, one should be $\lambda_1$ and another $\lambda_2$.

**Time Spent Reviewing:**

6

---

> ### Author Response · Authors · 2021-08-10
> **Responses to the comments on the optimization and others.**
>
> Thank you very much for your appreciation and constructive comments.  In the following, we provide detailed responses to your comments.
>
> *  **On the space of partitions in the optimization problems.** In both cases the space contains all the possible dyadic rectangles, i.e. rectangles obtained by a sequence of dyadic splits. For instance, for the first split there are $d$ possible splits.  A crude upper bound on the size of the space is the total number of sub-rectangles that $L_{d,n}$ has.  The size of the space in (4) and (8) is upper bounded by $N^2 =  n^{2d}$.   This is indeed exponential in terms of $d$.  In this paper, we consider fixed but arbitrary $d$ and in practice, we are not investigating moderately large $d$.
>
> * **On the algorithm.**  We follow the dynamic programming algorithm from the following paper
>
>    Donoho, D. L. (1997). CART and best-ortho-basis: a connection. The Annals of Statistics, 25(5), 1870-1911.
>
>    The algorithm finds the optimal solution among all dyadic partitions, and hence it is not a greedy algorithm.  The total complexity is
>    $O(n^d)$. We will add this comment to the manuscript if the paper is accepted.
>
> *  **On line 178--179, it is unclear what it means by  "instead of DCART, it is possible to deploy the ORT estimator [12] or the NP-hard estimator (2) in our algorithms".**
>
>     DCART considers dyadic partitions where in each step a rectangle can only be split by half along one of the $d$ coordinates. In contrast,
>     ORT picks one of the $d$ coordinates and it allows for the split to take place anywhere, not just in the middle. Hence, the ORT estimator
>     considers a much larger space and  it is more computationally-intensive than DCART. Our comment in lines 178--179 is to emphasize that
>      our theory also holds if we consider to analyze ORT instead of DCART.  The same holds for replacing DCART with the NP-hard estimator.
>
> * **On the necessary of additional merging step.**  You are correct that the penalty $\lambda_1 \vert \Pi\vert$ does discourage over-partitioning.  However, due to the fact that DCART only proceeds cuts in a restrictive way, which is not the $\ell_0$ penalty over all possible rectangles, we have shown both theoretically and numerically that this penalization _per se_ is not enough to eliminate all over-partitioning.
>
> * **In the merging stage, we merge, say, rectangles $R_i$ and $R_j$. From (9), they are close but they are not required to be neighbors. Is that a bit strange?**
>
>      This is a great question! what happens is that there could be a region of the lattice where the true signal takes on a constant value but the
>      algorithm separates it into two regions with small rectangles in the middle. If the rectangles that separate the region are small, the noise
>      can make it difficult for the merging step to correctly group the separated regions. So we find that it is better in practice and theory to
>      consider rectangles nearby and not only neighbors.  This allows for a better fit without adding much computational cost.  In fact, we have also considered more types of merging in our procedure than just neighboring rectangles, but this proposed merging allows for a better fit without adding much computational cost.

---

### Decision · Program_Chairs · 2021-09-28

**Decision:**

Accept (Poster)

**Comment:**

The paper provides an interesting framework for studying the problem of partition recovery for piecewise constant signals (in dimension >1). They propose a few modifications to the DCART algorithm and prove a few interesting theoretical results about their proposed algorithm. Given that the authors address the concerns of the reviewers satisfactorily, the paper should be accepted for presentation at Neurips.

**Consistency Experiment:**

NeurIPS has a long history of experimentation. In 2014, NeurIPS ran an experiment in which 10% of submissions were reviewed by two independent committees to quantify the randomness in the review process. This year, we repeated a variant of this experiment to see how the quality of the review process has changed over time.  This paper was part of the experiment and was therefore assigned to two committees (consisting of reviewers, an Area Chair, and a Senior Area Chair) that reached independent decisions.  If both committees made the same recommendation, this recommendation was followed. If a single committee recommended acceptance, the paper was accepted (with the exception of a few cases in which the other committee identified what we considered a fatal flaw, e.g., an error in a key result).

Both committees reached the same decision: **Accept (Poster)**

The other committee assigned to the paper recommended **Accept (Poster)**.  You can find the other set of reviews, along with any follow up discussion with the authors here:
https://openreview.net/forum?id=yITJ6t31eAE